# Sensory cortex plasticity supports auditory social learning

Nihaad Paraouty [1] ✉, Justin D. Yao[2], Léo Varnet [3], Chi-Ning Chou[4,5], SueYeon Chung[1,4] & Dan H. Sanes[1,6,7,8]

Social learning (SL) through experience with conspecifics can facilitate the acquisition of many behaviors. Thus, when Mongolian gerbils are exposed to a demonstrator performing an auditory discrimination task, their subsequent task acquisition is facilitated, even in the absence of visual cues. Here, we show that transient inactivation of auditory cortex (AC) during exposure caused a significant delay in task acquisition during the subsequent practice phase, suggesting that AC activity is necessary for SL. Moreover, social exposure induced an improvement in AC neuron sensitivity to auditory task cues. The magnitude of neural change during exposure correlated with task acquisition during practice. In contrast, exposure to only auditory task cues led to poorer neurometric and behavioral outcomes. Finally, social information during exposure was encoded in the AC of observer animals. Together, our results suggest that auditory SL is supported by AC neuron plasticity occurring during social exposure and prior to behavioral performance.

Social learning is a natural strategy that facilitates the acquisition of new behaviors. It typically occurs when a naïve animal experiences a conspecific performing a well-defined behavior and, as a consequence, acquires that behavior more rapidly than would occur without the social element[1–4]. Social learning is found throughout the animal kingdom, from bumblebees to primates, and can be mediated by all sensory modalities[5–15]. For example, gerbils and rats acquire dietary preferences by smelling or tasting food on the mouths of familiar conspecifics[16,17], chimpanzees and cockatoos learn tool use to obtain food by watching conspecifics[18,19], blue tits learn to associate a vocalization with antipredator behaviors by listening to heterospecific great tits[20], bats learn to avoid a poisonous toad through social observation of an acoustic prey cue[21], and human infants learn to speak through social exposure[22]. Perhaps the most well-studied form of auditory social learning is the acquisition of a species-specific vocalization by juvenile songbirds during exposure with an adult male tutor[23,24]. In each case, animals learn about sensory cues despite the absence of direct reinforcement, yet the

central plasticity mechanisms that support social learning remain uncertain.

The social learning paradigms for which neural mechanisms are well-studied implicate a broad range of structures through the use of correlational and causational experimental designs. For observational fear learning, a similar network of brain regions is thought to be involved both in direct fear learning and in social fear learning, including the anterior cingulate cortex and the amygdala[25–27]. Thus, local inactivation of the anterior cingulate cortex substantially impairs observational fear learning[26]. For the acquisition of motor behaviors through imitation, an analogous neural mechanism contributes to learning through observation or active practice[28,29]. Finally, social facilitation of song acquisition by juvenile songbirds involves both the anterior neostriatum and a higher auditory cortical region, the caudomedial nidopallium[30–34]. This auditory memory is required for the accurate production of songs that is generated subsequently through practice[35]. Therefore, social experience can improve the sensory encoding of environmental cues (i.e., neuronal responses become

[1]Center for Neural Science New York University, New York, NY 10003, USA. [2]Department of Otolaryngology, Rutgers University, New Brunswick, NJ 08901, USA. [3]Laboratoire des Systèmes Perceptifs, UMR 8248, Ecole Normale Supérieure, PSL University, Paris 75005, France. [4]Center for Computational Neuroscience, Flatiron Institute, Simons Foundation, New York, NY, USA. [5]School of Engineering & Applied Sciences, Harvard University, Cambridge, MA 02138, USA. [6]Department of Psychology, New York University, New York, NY 10003, USA. [7]Department of Biology, New York University, New York, NY 10003, USA. [8]Neuroscience Institute, NYU Langone Medical Center, New York, NY 10003, USA. ✉e-mail: np64@nyu.edu

more selective for relevant stimuli), and this initial plasticity may facilitate the acquisition of new behaviors.

Here, we test the hypothesis that social experience with a demonstrator gerbil performing a sound discrimination task induces neural plasticity in an observer gerbil that is causally related to the observer's enhanced rate of task acquisition. We previously showed that naïve observer gerbils can learn an arbitrary auditory discrimination task at a faster rate after exposure to a performing demonstrator gerbil, even in the absence of visual cues[36]. Unlike song acquisition in juvenile songbirds, the auditory discrimination task used here does not emerge naturally[37], and must hence be acquired de novo through either individual experience or social exposure. One plausible site of plasticity that could support this form of socially facilitated learning is the auditory cortex (AC). In fact, a compelling literature supports the direct involvement of AC in a range of learning paradigms[38–44], including those that rely on social experience[33,45].

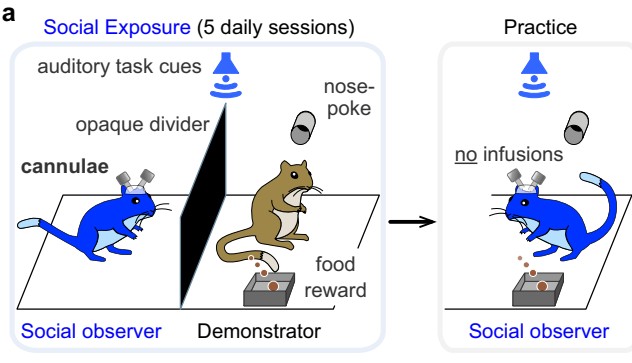

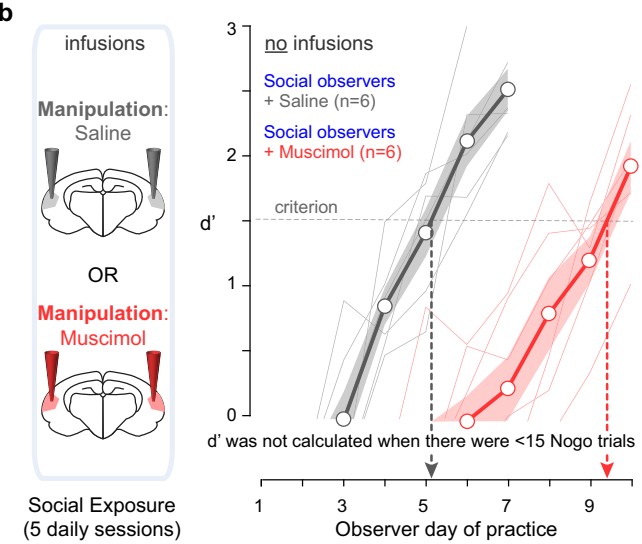

**Fig. 1 | Suppression of auditory cortex activity during social exposure delays task acquisition. a** Schematic of social learning paradigm. Left panel: A naïve *social observer* (blue) instrumented with bilateral cannulae in AC initially experiences five consecutive social exposure sessions with a trained demonstrator gerbil (brown) that is performing an AM discrimination task. The animals are separated by an opaque divider, preventing the *social observer* from having access to visual cues during exposure. Separate groups of *social observers* receive infusions of either saline or muscimol prior to each day of social exposure. Right panel: The *social observer* subsequently practices the AM discrimination task. There are no infusions prior to practice days. **b** Task acquisition during practice sessions is assessed with the signal detection metric, *d′*, and performance is plotted as a function of the day of practice. Criterion *d′* set at 1.5. No *d′* was computed when the *social observers* initiated <15 Nogo trials in the practice sessions. Thin lines denote individual animals; thick lines and transparent areas denote mean ± SE. Muscimol infusion caused a significant delay in social learning (Steel–Dwass nonparametric comparison, two-sided, *p* = 0.033).

Therefore, we sought to test two predictions that emerge from the hypothesis: first, that inactivating AC during social exposure would diminish social learning, and second, that AC sensitivity to auditory task cues would be enhanced by social exposure.

## Results

### The auditory cortex is necessary for social learning

We first asked whether there was a causal relationship between AC activity and auditory social learning. To address this question, we implanted the AC of *social observer* animals bilaterally with cannulae for the purpose of infusing muscimol, a GABA$_A$ receptor agonist, to depress cortical activity (see Methods). As described previously[36,46], demonstrator gerbils were trained by the experimenter to perform a Go-Nogo amplitude modulation (AM) rate discrimination task. Briefly, the demonstrators were placed on controlled food access and trained to initiate each trial by placing their noses in a nose port. The Go stimulus (12 Hz AM noise, 100% depth) indicated the presence of a reward at the food tray, while the Nogo stimulus (4 Hz AM noise, 100% depth) signaled the absence of a food reward. A discrimination performance metric, d-prime (*d′*), was calculated for each session as $d′ = z$(hit rate) – $z$(false alarm rate). To qualify as a demonstrator, animals were required to perform the task with a $d′ > 1.5$ (see Methods).

A trained and performing demonstrator gerbil was paired with a naïve *social observer* for five consecutive daily exposure sessions (Fig. 1a). The demonstrator compartment was separated from the *social observer* compartment by an opaque divider to prevent access to visual cues. However, the *social observer* had access to all other social cues (i.e., vocalizations, pellet chewing, and movement-generated sounds), as well as all task-specific sound cues (i.e., the Go and Nogo sound stimuli delivered from above the test cage). During each exposure session, the *social observer* experienced at least 80 Go trials (min–max = 82–184) and 20 Nogo trials (24–59) performed by the demonstrator. Immediately following the fifth exposure session, the demonstrator was removed, as well as the divider and the *social observer* were permitted to practice the task on their own (Fig. 1a, see also Supplementary Fig. 1). The sensitivity metric, *d′*, was computed for all sessions during which the *social observer* performed ≥15 Nogo trials.

Before each of the five exposure sessions, the cannulae-implanted *social observers* were briefly anesthetized and received bilateral AC infusions of either saline or muscimol (see Methods). Animals were allowed to recover for 15–20 min before the start of the exposure session. No infusions of saline or muscimol were delivered prior to the practice sessions. Anatomically confirmed cannulae tracks are shown in Supplementary Fig. 2a.

On average, the *social observers* who received saline infusions during the exposure sessions (*n* = 6) required 5.2 ± 0.3 days (mean ± standard error) to perform the task at a criterion *d′* of 1.5 (Fig. 1b, gray lines). No significant difference was found between the number of days taken to reach the criterion *d′* by the current saline-infused *social observers* and those tested without any infusions or cannulae implants in our previous report (see Fig. 3 [36]; Steel–Dwass nonparametric comparison, *p* = 0.967). This suggests that neither the surgery nor the saline infusions under anesthesia prior to the exposure sessions impacted the subsequent rate of task acquisition during the practice sessions.

To test whether AC inactivation could significantly delay the subsequent task acquisition during the practice sessions, a subset of *social observers* (*n* = 6) received muscimol infusions prior to each of the 5 exposure sessions. No muscimol was infused during the subsequent practice sessions. The muscimol-infused *social observers* required 9.4 ± 0.4 days to perform the task at a criterion *d′* of 1.5 (Fig. 1b, red lines). The rate of task acquisition was significantly delayed for the muscimol-infused *social observers* as compared to the saline-infused *social observers* (Steel–Dwass nonparametric comparison, *p* = 0.033). Although we cannot rule out a long-term effect of muscimol on task

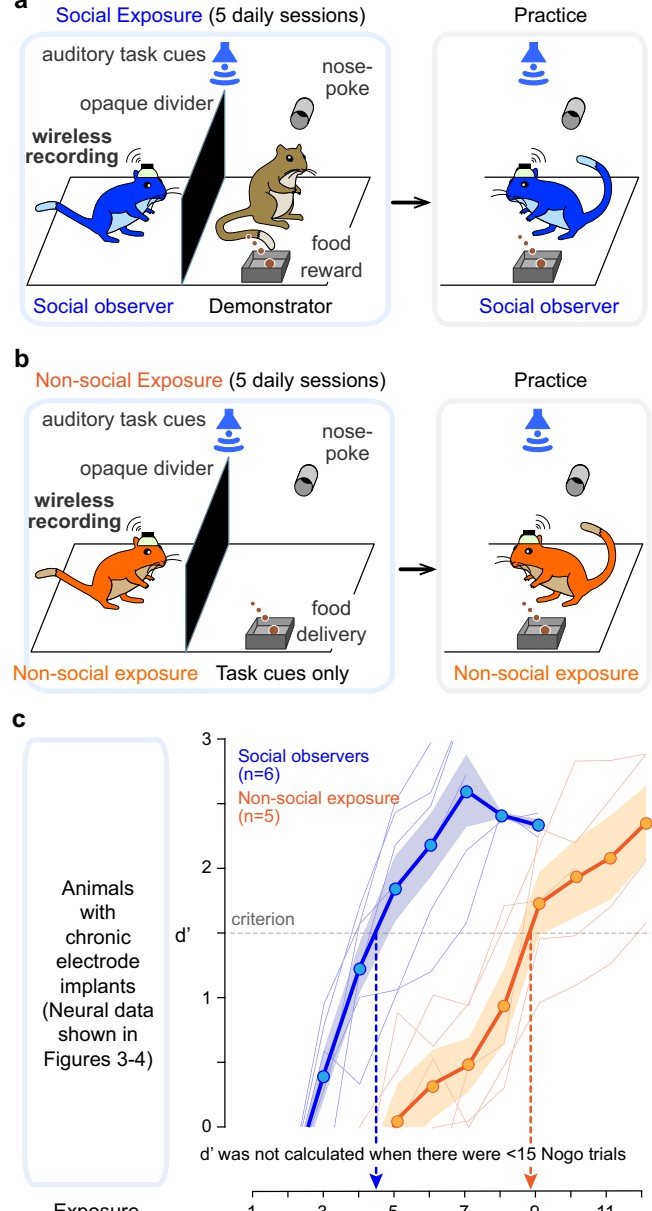

**Fig. 2 | Social exposure enhances the rate of task acquisition. a** Schematic of the social learning paradigm. Left panel: A naïve *social observer* (blue) instrumented with an electrode array implanted in AC initially experiences five consecutive social exposure sessions with a trained demonstrator gerbil (brown) that is performing an AM discrimination task. Right panel: The *social observer* subsequently practices the AM discrimination task. **b** Schematic of the non-social exposure paradigm. Left panel: A naïve *non-social exposure animal* (orange) instrumented with an electrode array implanted in AC initially experiences five consecutive non-social exposure sessions to experimenter-triggered auditory task cues. Right panel: The *non-social exposure* animals subsequently practice the AM discrimination task. **c** Left panel: AC recordings were obtained during each exposure day from both *social observers* and *non-social exposure* animals. Right panel: Task acquisition during practice sessions was assessed, and behavioral *d′* of *social observers* and *non-social exposure* animals is plotted as a function of the day of practice. No *d′* was computed when observers initiated <15 Nogo trials in the practice sessions. Thin lines denote individual animals; thick lines and transparent areas denote mean ± SE. *Social observers* reached criterion *d′* in significantly fewer days than *non-social exposure* animals (Steel–Dwass non-parametric comparison, two-sided, *p* = 0.037).

acquisition, previous results[39] indicate an absence of long-term muscimol effects on psychometric performance. Hence, our results show that AC may contribute to auditory social learning and suggest that it may store task-specific acoustic information during the exposure phase.

To evaluate whether the effect of muscimol infusion persisted into the first 1 or 2 days of practice, we excluded these days from the data analysis and confirmed a significant delay in the rate of task acquisition, as compared to the saline-infused *social observers* (after excluding practice day 1, *p* = 0.033; after excluding practice days 1 and 2, *p* = 0.049). Furthermore, to determine whether muscimol-infused *social observers* benefited from the social exposure sessions, we compared their learning curves to previously collected control groups in which there was no social experience[36]. These comparisons revealed that muscimol-infused *social observers* continued to display an effect of *social exposure* as they performed significantly better than all previously obtained control groups (comparison with Control for social exposure from previous report[36], Fig. 2a: no animal reached criterion *d′*; comparison with Control for cage exposure from Fig. 2b: *p* = 0.049; comparison with Control for all exposure from Fig. 2c: *p* = 0.011).

### Social exposure leads to faster task acquisition as compared to non-social exposure

To assess the neural mechanisms underlying auditory social learning, another group of naïve *social observers* (*n* = 6) were chronically implanted with an electrode array in the left AC (see Methods) and paired with a demonstrator gerbil for 5 consecutive exposure sessions, as described above (Fig. 2a). On average, the electrode-implanted *social observers* required 5.0 ± 0.5 days to perform the task at a criterion *d′* of 1.5 (Fig. 2c, blue lines, see also Supplementary Fig. 3). No significant difference was found between the number of days taken to reach the criterion *d′* by the current implanted *social observers* and those tested without any implants in our previous report[36] (Fig. 3; Steel–Dwass comparison; *p* = 0.999; Supplementary Fig. 4). This confirms that the electrode implant surgery did not impact the rate of learning.

To control for social experience, a second group of naïve gerbils was chronically implanted with an electrode array in the left AC but received only non-social exposure to experimenter-triggered auditory task cues (*n* = 5; Fig. 2b). These cues included the task sound stimuli (i.e., Go and Nogo sound stimuli delivered from above the test cage), and the response contingencies (i.e., pellet delivery to simulate Hits following a Go stimulus, time-outs to simulate False Alarms following a Nogo stimulus, as well as Misses and Correct Rejects). In fact, the experimenter-triggered exposure sessions matched the Go and Nogo trials of actual demonstrator animals (from Fig. 2a; see Methods). Following the five daily exposure sessions, the *non-social exposure* animals were permitted to practice the task. The *non-social exposure* animals required 9.6 ± 0.8 days to learn the task at the criterion *d′* of 1.5 (Fig. 2c, orange lines; see also Supplementary Fig. 3). No significant difference was found between the number of days taken to reach the criterion *d′* by the current implanted *non-social exposure* animals and those tested previously[36] (Fig. 4; Steel–Dwass comparison; *p* = 0.934; see Supplementary Fig. 4). However, the implanted *non-social exposure* animals took significantly longer as compared to the implanted *social observers* to reach criterion *d′* (*p* = 0.037). Thus, the addition of social cues during exposure sessions significantly improves the rate of task acquisition during practice.

### Social exposure changes AC neuron response properties to auditory task cues

To determine whether social exposure altered the basic response properties of AC neurons to auditory task cues, we analyzed auditory-driven AC responses during three distinct epochs: (1) an early exposure period, consisting of recordings during exposure days 1–2, (2) a late

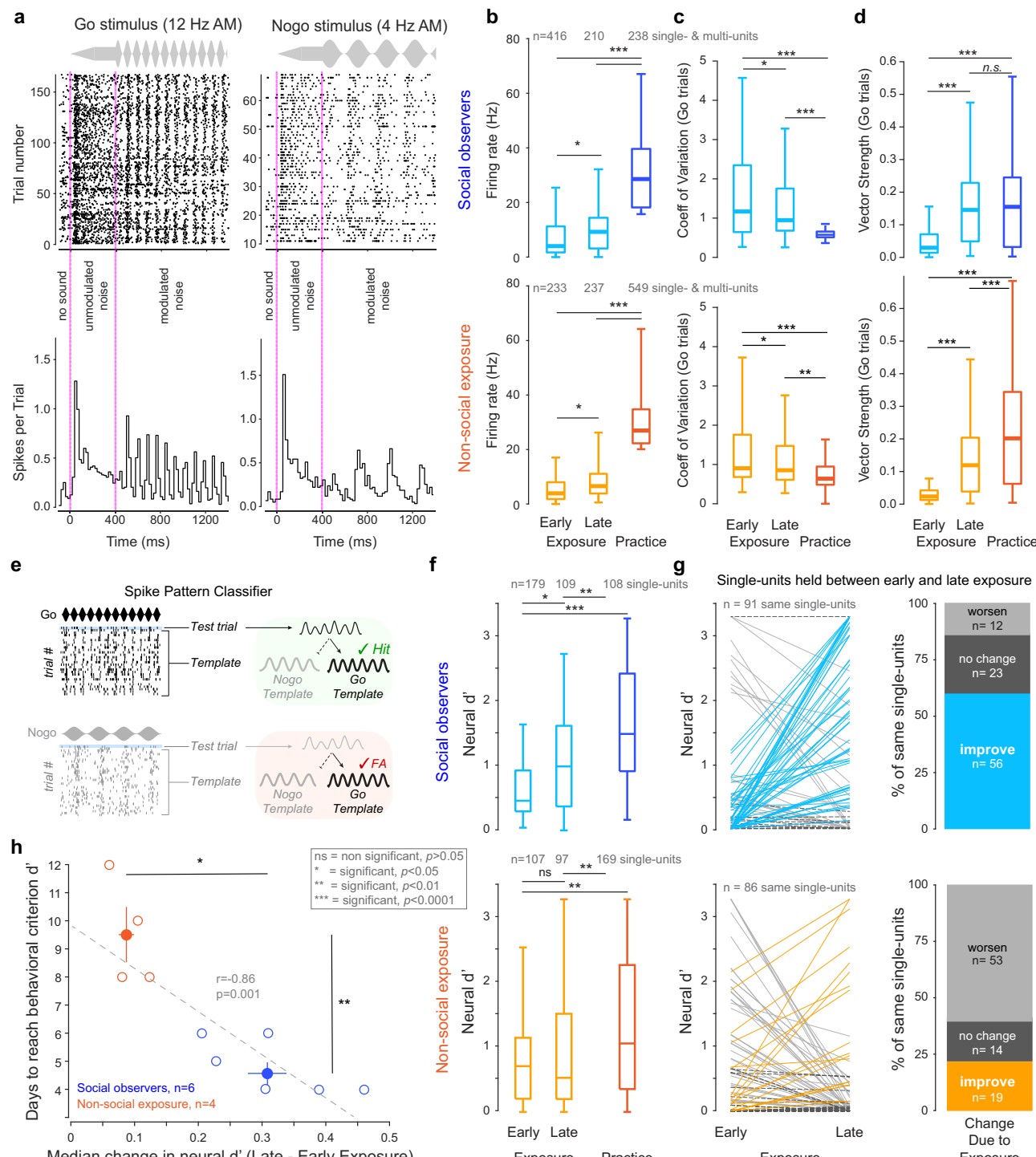

**Fig. 3 | Social exposure-induced auditory cortex plasticity correlates with the rate of task acquisition. a** Raster and PSTH of example AC cell to Go and Nogo stimuli during social exposure. **b** Firing rate of single- and multi-units to modulated AM portions of stimuli (central line: median, bottom and top edges: 25th and 75th percentiles, whiskers: minimum and maximum points). Significant increases in FR from early to late exposure to practice were observed for both groups. **c** For both groups, the coefficient of variation (CV) of single and multi-units decreased significantly from early to late exposure to practice. **d** For both groups, vector strength (VS) of single and multi-units improved significantly from early to late exposure, and for *non-social exposure* animals only, VS also increased from late exposure to practice. See text for statistics (Steel−Dwass comparisons, two-sided). **e** Schematic of spike pattern classifier analysis to calculate neural *d'* values for

single units. **f** Neural *d'* of AC single-units: *social observers* displayed a significant increase from early to late exposure, but *non-social exposure* animals did not (Steel−Dwass comparisons, two-sided). **g** Mean neural *d'* are plotted for a subset of putatively same single-units held from early to late exposure. Significant group differences are found with exposure (Likelihood ratio Chi-square, *p* = 0.0006). **h** Significant correlation between the difference in median neural *d'* from early to late exposure of all single units for each animal and the number of practice days required for the given animal to perform at criterion *d'*. Significant differences were observed between the two groups in terms of number of practice days to reach criterion *d'* (Wilcoxon rank sum test, two-sided, $X^2(1) = 6.79$, $p = 0.009$) and median neural *d'* change during exposure ($X^2(1) = 6.56$, $p = 0.011$).

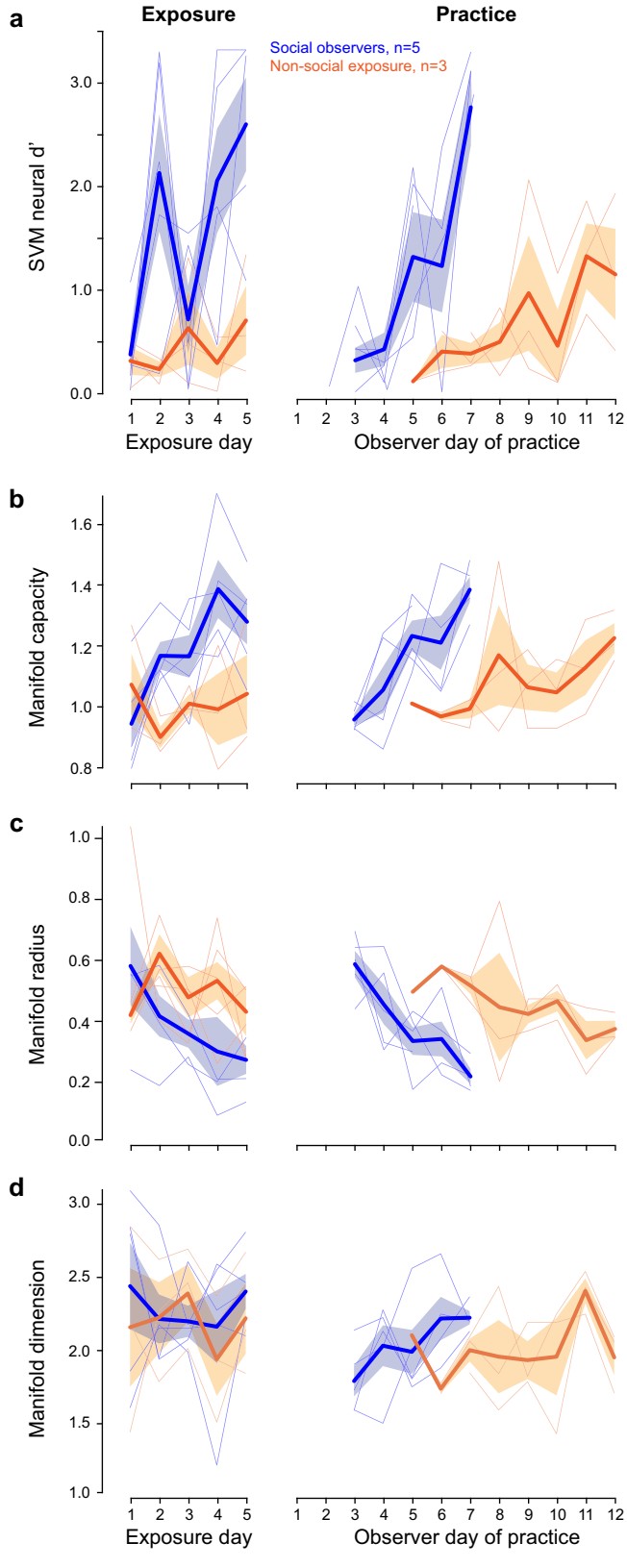

**Fig. 4 | Auditory cortex population neural sensitivity improves during social exposure. a** Left panel: AC population neural $d'$ during exposure for *social observers* (blue lines) and *non-social exposure* animals (orange lines). Thin lines denote individual animals; thick lines and transparent areas denote mean ± SE. Right panel: AC population neural $d'$ during practice. A significant group difference, a significant effect of days, and a significant interaction were observed between the group and days (mixed-model ANOVA; group: $p < 0.0001$; days: $p < 0.0001$; interaction: $p = 0.014$). **b** Left panel: Manifold capacity values during exposure. Right panel: Manifold capacity values during practice. A significant group difference, a significant effect of days, and a significant interaction were observed between the group and days (group: $p < 0.0001$; days: $p = 0.0001$; interaction: $p = 0.0001$). **c** Left panel: Manifold radius values during exposure. Right panel: Manifold radius values during practice. A significant group difference, a significant effect of days, and a significant interaction were observed between the group and days (group: $p = 0.001$; days: $p = 0.0001$; interaction: $p = 0.008$). **d** Left panel: Manifold dimension values during exposure. Right panel: Manifold dimension values during practice. No significant group difference nor a significant effect of days was observed (group: $p = 0.277$; days: $p = 0.425$).

tracks within the AC are shown for two representative implanted animals in Supplementary Fig. 2b. Example raster plots for a single unit recorded during social exposure are shown in Fig. 3a, in response to the Go stimulus (12 Hz AM) and the Nogo stimulus (4 Hz AM).

For both groups, the firing rate (FR) evoked by the modulated portion of the sound stimuli (that is, the AM) increased significantly from early to late exposure (Fig. 3b, *social observers*, Tukey–Kramer HSD comparisons; $p = 0.016$; *non-social exposure*, $p = 0.041$). The FR responses during practice for both groups were significantly higher, as compared to the exposure epochs (both $p < 0.0001$). Overall, no significant FR difference was observed between *social observers* and *non-social exposure* animals for responses to the AM stimuli during both exposure and practice (mixed-model ANOVA, $F(1,2) = 0.656$, $p = 0.4182$). No significant changes in FR from early to late exposure were found either during silence or during the unmodulated portion of the sound stimulus (see Supplementary Fig. 5, $p > 0.05$).

Next, we examined FR variability using the coefficient of variation (CV) and found a significant decrease in CV for both groups (Fig. 3c; Kruskal–Wallis $H$ test, $X^2(2) = 171.04$, $p < 0.0001$). For *social observers*, the decrease in CV occurred from early (median $= 1.17 \pm 1.61$) to late exposure ($0.95 \pm 1.5$; Steel–Dwass comparison, $p = 0.029$) and from late exposure to practice ($0.58 \pm 0.12$; $p < 0.0001$). *Non-social exposure* animals displayed a similar effect with exposure (early, $0.9 \pm 1.15$; late, $0.85 \pm 1.01$; $p = 0.0308$) and practice ($0.64 \pm 0.75$; $p = 0.0005$). Finally, we examined vector strength (VS), a measure of phase-locking. For *social observers*, we found a significant improvement in VS from early to late exposure (Fig. 3d; early, median $= 0.07 \pm 0.13$; late, $0.16 \pm 0.14$; Steel–Dwass comparison, $p < 0.0001$) but no significant change from late exposure to practice ($0.17 \pm 0.15$; $p = 0.826$). For *non-social exposure* animals, we found a similar significant increase in VS from early to late exposure (early, $0.04 \pm 0.09$; late, $0.14 \pm 0.12$; $p < 0.0001$) and from late exposure to practice ($0.22 \pm 0.18$; $p < 0.0001$). Although there is a trend, no significant group difference was found during early exposure ($p = 0.083$) and during practice ($p = 0.055$). However, during late exposure, the *social observers* had significantly higher VS as compared to the *non-social exposure* animals ($p = 0.038$). Together, these results suggest that both social and non-social exposure are associated with changes in AC neuron FR and temporal processing. However, they do not tell us whether these changes lead to improved stimulus discriminability. This is precisely what we sought to address next.

## Neural sensitivity improves during social exposure and predicts behavioral performance

Since the ability of AC neurons to discriminate between Go and Nogo stimuli could facilitate task acquisition, we computed a measure of

exposure period, consisting of recordings during exposure days 4–5, and (3) a practice period, consisting of recordings during all practice days. A total of 864 AC units (396 single-units) were recorded from *social observers* (exposed to a performing demonstrator animal; $n = 6$), and a total of 1019 AC units (373 single-units) were recorded from *non-social exposure* animals (exposed to only experimenter-triggered auditory task cues; $n = 5$ animals). Anatomically confirmed electrode

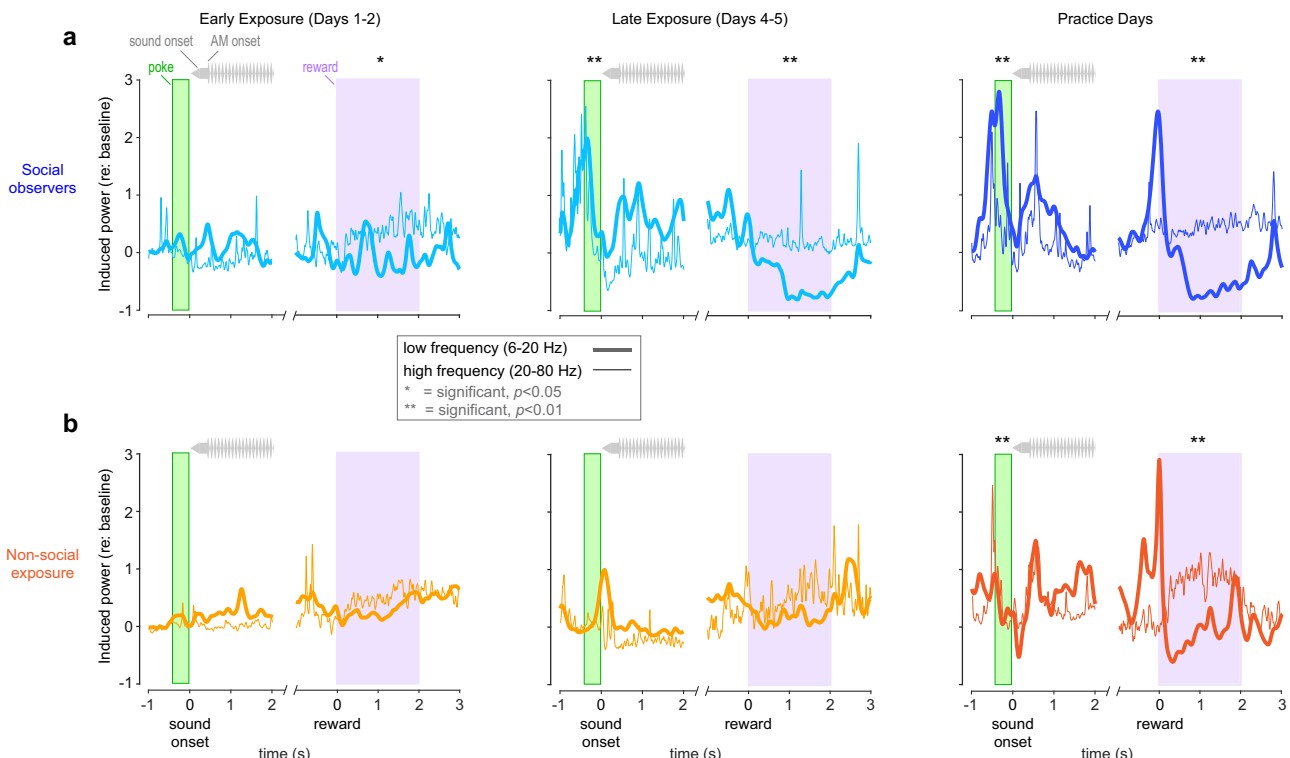

**Fig. 5 | Social observer auditory cortex encodes demonstrator behavioral cues.** **a** *Social observer* mean induced power fluctuations in response to Hit trials during early exposure (left), late exposure (middle), and practice (right, only practice sessions with *d′* > 1.5 were included). Traces are aligned with demonstrator nose-poke (green bar) and demonstrator reward delivery (purple bar). Induced power (i.e., oscillatory activity not necessarily time-locked to the AM signals) is shown for 2 frequency bands: 6–20 Hz (thick lines) and 20–80 Hz (thin lines). **b** *Non-social exposure* animal mean induced power fluctuations in response to Hit trials during early exposure (left), late exposure (middle), and practice (right). For *social*

*observers*, there was a significant increase in induced power during the demonstrator nose-poke for late exposure and practice sessions (corrected two sample *t*-tests for individual animals; see Supplementary Table 1). However, these increases in induced power were only observed during practice for *non-social exposure* animals. For *social observers*, there was also a significant dip in induced power during reward delivery for early exposure, late exposure, and practice sessions. However, this decrease in induced power was only observed during practice for *non-social exposure* animals.

neural sensitivity. A signal detection theory metric, neural *d′*, was calculated for each AC unit using a pattern classifier analysis (Fig. 3e). This analysis was restricted to single units and used Euclidean distance to quantify the dissimilarity between two sets of spike trains in high-dimensional space (see Methods).

Figure 3f summarizes *d′* values for all single AC neurons recorded during exposure and practice sessions. For *social observers*, we found a significant improvement in neural *d′* from early (median = 0.47 ± 0.44) to late exposure (1.00 ± 0.77; Steel–Dwass comparison, *p* = 0.029). Neural *d′* values further improved from late exposure to practice (1.50 ± 1.07; *p* = 0.006). In contrast, *non-social exposure* animals did not display a significant change in neural *d′* from early to late exposure (early, 0.71 ± 0.61; late, 0.53 ± 0.98; *p* = 0.8835), but a significant improvement was present for practice (1.07 ± 1.08; *p* = 0.003). Together, these results indicate that only social experience led to an enhancement of neural discriminability during the exposure sessions. In contrast, single units from both *social observers* and *non-social exposure* animals showed a significant enhancement in neural discriminability during practice.

To further assess whether the sensitivity of individual AC neurons improved during exposure, we restricted our analysis to a subset of single units obtained from identical recording sites with similar waveforms and spontaneous FRs (±2 Hz) during both early and late exposure sessions (see Methods). First, we generated 10 unique values of neural *d′* for each individual cell as the spike pattern classifier analysis uses one randomly selected spike train from all Go trials as the Go Test trial and one randomly selected spike train from all Nogo trials as the Nogo Test trial. Next, we computed two-tailed *t*-

tests to examine whether the changes across exposure were significant for each AC neuron that was putatively held from early to late exposure sessions. Figure 3g (left panel) shows the mean *d′* values for each AC single-unit during early and late exposure. For the *social observers*, 62% of single units (*n* = 56; Fig. 3g, top right panel) displayed a significant improvement in neural *d′* from early to late exposure. In contrast, for the *non-social exposure* animals, only 22% of single units (*n* = 19; Fig. 3g, bottom right panel) displayed a significant increase in neural *d′*. Furthermore, a significant difference was found between the two groups of animals in terms of the distribution of cells (i.e., cells that significantly improved or worsened or those without significant change; Likelihood ratio Chi-square, *p* = 0.0006).

If the exposure-induced improvements in neural sensitivity of single AC units are associated with faster task acquisition during practice, then we would expect to see a correlation between these two measures. Figure 3h shows the median improvement in neural *d′* with exposure for each animal plotted against the number of practice days required for that animal to perform the task at criterion *d′*. We found a significant correlation (Pearson's, *p* = 0.001), indicating that improvements in AC neuron sensitivity during exposure accurately predicted subsequent task acquisition during practice. Although a significant correlation is found for all animals (both *social observers* and *non-social exposure* animals), the exact relationship between median improvement in neural *d′* from early to late exposure and the number of days to perform at criterion *d′* may be different for the two groups (*social observers*, *n* = 6, *p* = 0.02; *non-social exposure* animals, *n* = 4, *p* = 0.14).

## Population neurometrics improve during social exposure and practice

Population coding offers a global neural readout for individual animals and may, in turn, provide a more direct comparison to behavioral performance. First, using linear classifiers based on support vector machines[47] (SVM, see Methods), we computed a population neural $d'$ for simultaneously recorded single-units in each session (Fig. 4a). The underlying assumption of this approach is that it informs how a decoder, downstream of the AC, could use the information represented by the AC population. A mixed-model ANOVA revealed a significant group difference ($F(1,78) = 25.99$, $p < 0.0001$), and a significant effect of days ($F(1,78) = 25.78$, $p < 0.0001$). A significant interaction was present between groups and days ($F(1,78) = 6.29$, $p = 0.014$): the *social observers* showed significantly larger population neural $d'$ improvement as compared to the *non-social exposure* animals. In fact, *social observers* displayed a significant effect of days during the 5 daily exposure sessions ($F(1,23) = 7.94$, $p = 0.0098$), as well as during practice ($F(1,23) = 18.93$, $p = 0.0002$). In contrast, *non-social exposure* animals did not display a significant effect of days during exposure ($F(1,13) = 1.34$, $p = 0.268$), although a significant improvement was present during practice ($F(1,19) = 11.70$, $p = 0.003$).

To assess the high-dimensional geometry of the population code, we examined a second population metric based on the mean-field theoretic manifold analysis technique that connects stimulus encoding efficiency to decoding capacity[48–50] (see Methods). For example, a large manifold capacity suggests efficient stimulus representation (i.e., high separability of manifolds). Thus, the capacity measure of Go and Nogo manifolds (i.e., point-cloud manifolds from single unit spike count data) were obtained for all animals (Fig. 4b). A mixed-model ANOVA revealed a significant group difference ($F(1,78) = 21.03$, $p < 0.0001$), and a significant effect of days ($F(1,78) = 16.74$, $p = 0.0001$). A significant interaction was present between groups and days ($F(1,78) = 10.96$, $p = 0.001$): the *social observers* showed significantly larger manifold capacity improvement as compared to the *non-social exposure* animals. Furthermore, *social observers* displayed a significant effect of days during both exposures ($F(1,23) = 13.06$, $p = 0.002$) and practice sessions ($F(1,23) = 17.64$, $p = 0.0003$). In contrast, the *non-social exposure* animals did not display a significant effect of days during exposure ($F(1,13) = 0.01$, $p = 0.914$), but a significant improvement was present during practice ($F(1,19) = 4.69$, $p = 0.043$).

In this framework, geometric measures such as radius and dimensionality inform us about the encoding of high-level features, including the extent and shape of neural variability in response to the same stimulus[48]. Our analyses indicated that the overall change in manifold capacity was related to a decrease in manifold anchor radius rather than to a change in dimension. The manifold radius (Fig. 4c) decreased significantly with days ($F(1,78) = 16.41$, $p = 0.0001$) and was significantly smaller for the *social observers* as compared to the *non-social exposure* animals ($F(1,78) = 11.15$, $p = 0.001$). A significant interaction between groups and days was also present ($F(1,78) = 7.34$, $p = 0.008$). In contrast, the manifold dimension (Fig. 4d) was similar across groups ($F(1,78) = 1.21$, $p = 0.277$) and remained unchanged with days ($F(1,78) = 6.88$, $p = 0.425$).

Neural population $d'$ values during exposure were significantly correlated with behavioral measures of $d'$ during practice for all animals (Supplementary Fig. 6a; Pearson's; $p = 0.002$). Similarly, there was a correlation between manifold capacity (Supplementary Fig. 6b; $p = 0.002$) and manifold radius during exposure and the subsequent behavioral $d'$ during practice (Supplementary Fig. 6c; $p = 0.012$). In contrast, the manifold dimension during exposure did not display a correlation with behavioral $d'$ during practice (Supplementary Fig. 6d; $p = 0.43$).

In addition, behavioral measures of $d'$ during practice were significantly correlated with neural population $d'$ (Supplementary Fig. 6e; $p < 0.0001$), manifold capacity (Supplementary Fig. 6f; $p = 0.007$), manifold radius (Supplementary Fig. 6g; $p = 0.0007$), and manifold dimension during practice (Supplementary Fig. 6h; $p = 0.007$).

To summarize, during the exposure sessions, the population neurometric measures obtained from *social observers* displayed a significant improvement, while those from the *non-social exposure* group did not. This suggests that social cues may also be encoded by AC cell populations and could be pivotal to the enhanced representation of auditory task cues, as well as the behavioral advantage that is observed during practice for *social observers*.

## AC represents social cues from the demonstrator during exposure sessions

To address whether AC activity was modulated by social cues from the demonstrator during the exposure sessions, we examined the correlation between local field potentials (LFP) and two demonstrator behaviors: nose-poking to initiate trials and acquisition of a reward at the food tray. First, we measured the induced power from LFP (see Methods) in a subset of active recording sites for each animal. Induced power represents an oscillatory activity that is not necessarily time-locked to the AM signals and is often related to a variety of higher-order cognitive functions, including internally driven representations[51,52]. Thus, induced power may reflect the encoding of social cues linked to the demonstrator's behavior during exposure sessions.

Our analysis targeted Go and hit trials and focused on 2 time periods: (1) an anticipatory phase during which demonstrators nose-poked to initiate each trial, just prior to the onset of the sound stimuli: from −0.5 to −0.1 sec (with 0 s representing sound onset), and (2) a reward-related phase during which the demonstrator sought and obtained a food reward: from 0 to 2 s (with 0 s representing reward onset). We examined induced power during those 2 time periods and looked at two distinct frequency bands: a low-frequency band (Fig. 5; 6–20 Hz; thick lines) and a high-frequency band (20–80 Hz; thin lines).

During early exposure (Fig. 5a, b, left panels), no significant induced power was observed during the anticipatory period, either for *social observers* or *non-social exposure* animals (two-tailed $t$-tests for each individual animal, $p > 0.05$, see Supplementary Table 1 for exact $p$ values). However, during late exposure, a significant increase of induced power was observed during the anticipatory period for *social observers* in both frequency bands but not for *non-social exposure* animals (for each individual *social observer,* $p < 0.05$; for each individual *non-social exposure* animal, $p > 0.05$). This increase in anticipatory power was observed during practice for all animals (all $p < 0.05$).

We also found a significant decrease in the low-frequency band for *social observers* when the demonstrators obtained a food reward on hit trials (Fig. 5a). This activity was present during both early and late exposure, as well as during practice for all *social observers* (all $p < 0.05$; see Supplementary Table 1 for exact $p$ values). For *non-social exposure* animals, we did not observe a significant decrease in low-frequency power related to reward delivery during exposure (all $p > 0.05$). To summarize, AC activity in *social observers* was closely linked to the *demonstrator's* behavior during exposure, suggesting that AC integrates both acoustic and social cues related to the demonstrator's general behavior.

For *social observers*, during practice, a significant correlation was found between the average power of the low-frequency band during the nose-poking period and the animal's behavioral $d'$ (Pearson's, $r = 0.31$, $p = 0.037$). In contrast, no significant correlation was found for the high-frequency band ($r = 0.09$; $p = 0.563$). For the *non-social exposure* animals, during practice, no significant correlation was found between the average power during the nose-poking period and the animal's behavioral $d'$ ($r = 0.15$, $p = 0.466$; $r = 0.12$; $p = 0.615$ for low and high-frequency bands, respectively). For both groups, no significant correlation was found between the average decrease in power during the reward period and the animal's behavioral $d'$ (for *social*

*observers*: $r = 0.07$, $p = 0.724$; $r = 0.04$, $p = 0.855$ for low and high-frequency bands, respectively; for *non-social exposure* animals: $r = 0.03$; $p = 0.890$; $r = 0.09$; $p = 0.563$). Together, these results indicate that only the induced LFP activity during nose-poke can accurately predict behavioral performance during practice sessions for *social observers*.

Finally, we examined the LFP-evoked power in response to auditory task cues (Supplementary Fig. 7). Stimulus-evoked power measures the degree of precise phase-locking of AC activity to the auditory AM stimuli across trials (see Methods). We observed significant changes from baseline at sound onset during both late exposure and practice for all *social observers* (two-tailed *t*-tests for each individual animal, $p > 0.05$; see Supplementary Table 2 for exact $p$ values). Significant onset responses were found for *non-social exposure* animals during practice only (all $p < 0.05$). We then computed a fast-Fourier transform (FFT, see Methods) and confirmed peak RMS power at 12 and 4 Hz for Go and Nogo trials, respectively (Supplementary Fig. 7c). The mean peak FFT amplitude increased significantly with exposure for both *social observers* and *non-social exposure* animals (mixed-model ANOVA, early vs. late exposure; $F(1,36) = 39.9$, $p < 0.0001$). No significant group effect was present (*social observers* vs. *non-social exposure* animals; $F(1,36) = 3.3$, $p = 0.078$), but a significant interaction between group and exposure period was found ($F(1,36) = 4.5$, $p = 0.039$), revealing larger FFT peak enhancement from early to late exposure for *social observers* as compared to *non-social exposure* animals. These results are consistent with our findings from AC single unit and AC population in terms of the encoding of auditory task cues.

## Discussion

The acquisition of a wide range of complex behaviors or skills can be facilitated by a period of social experience with a performing demonstrator, suggesting that a common set of neural mechanisms may contribute to both social learning without explicit reward and reinforcement learning through active practice[53]. One possibility is that environmental cues present during social experience are stored in sensory cortices and may later be exploited to facilitate the acquisition of the new behavior. To test this idea, we used a behavioral paradigm in which naïve gerbils acquire an auditory discrimination task more rapidly after a social exposure period with a performing demonstrator, even in the absence of visual cues[36]. First, we asked whether AC activity was required for this form of social learning, despite the absence of explicit reinforcement, in line with the causal role of AC in other forms of auditory learning[38,39,54]. We found that bilateral suppression of AC activity during social exposure resulted in a significant learning delay during the subsequent practice phase (Fig. 1). Next, we asked whether the neural representation of auditory task cues was remodeled by social and/or non-social exposure. Our results showed that both types of exposure altered basic AC response properties to auditory task-relevant cues (Fig. 3b–d). However, only social exposure led to an improvement in AC single neuron and AC population sensitivity to the auditory task cues, as compared to non-social exposure (Fig. 3g, h, Fig. 4a–c). While many forms of auditory learning are associated with AC plasticity[40–44,55–58], neural changes have principally been associated with practice. In contrast, our findings demonstrate that AC plasticity is selectively enhanced prior to active engagement in the task during the period of social exposure to a performing demonstrator.

The enhancement of neural discriminability induced by exposure correlated significantly with the subsequent rate of task acquisition during practice (Fig. 3h). This suggests that social context facilitates AC neuron storage of task-relevant information and may represent a form of incidental learning[59,60]. This form of AC plasticity resembles hippocampal place cell remodeling that occurs as animals actively explore their environment without any specific task. For example, following repeated exposure to two different environments, rat CA1 place cells acquire specific representations of each environment

despite the absence of explicit reinforcement[61]. Similarly, following repeated passive exposure to novel vocalizations, zebra finch caudo-medial nidopallium neurons display improved vocalization discriminability that lasted for at least 20 hours[62].

Enhanced neural sensitivity during social exposure may, in turn, improve the saliency of information relayed to downstream AC processing areas[63–68]. For example, auditory decisions involve both non-cortical and cortical regions downstream of the AC, such as the striatum and parietal cortex[69–75]. Therefore, we used a linear classifier to ask whether improvements in AC population representation could support enhancement in sensory discriminability. During both exposure and practice, AC population sensitivity improved for *social observers* but only improved during practice for *non-social exposure* animals (Fig. 4a).

To probe the representation-level structure of learning, we applied an analytical framework that connects stimulus encoding efficiency to decoding capacity[48–50]. During both exposure and practice, the classification capacity improved for *social observers* but only improved during practice for *non-social exposure* animals (Fig. 4b). Therefore, social exposure drives an improvement in decoding capacity. The manifold radius, which represents the extent of neural variability corresponding to the same stimulus, offers further insight. Manifold radius changes over time show that exposure sessions improve stimulus encoding efficiency by reducing neural variability for *social observers* (i.e., decrease in radius), which does not occur for *non-social exposure* animals. Interestingly, exposure did not change the dimensionality of the manifolds for either *social observers* or *non-social exposure* animals (i.e., there was no significant reduction in the shape of neural variability during both types of exposure). The fact that the extent of neural variability (radius) reduces, while the shape of this variability (dimension) does not, points to a potential population-level mechanism underlying the higher decoding capacity of *social observers* during exposure.

If the improvement in AC neuron sensitivity during exposure reflects a memory that is used during practice, then neural $d'$ would be expected to remain high at the onset of practice for *social observers*. In contrast, we found a decrease in neural $d'$ at the onset of practice (Fig. 4). One plausible explanation is that AC population network plasticity acquired during exposure is present but is not initially observed during task performance because it must first become associated with the top-down circuitry that supports task acquisition. Thus, the enhanced neural sensitivity (i.e., the ability of AC neurons to discriminate between the Go and Nogo stimuli) induced by social exposure would have to be integrated with new information about task initiation (i.e., nose-poking to initiate sounds) and task structure (i.e., availability of reward following Go stimuli in a limited time window, and avoidance of food tray following Nogo stimuli to prevent time-outs). Such task-dependent modulation of neural coding properties is common. For example, a large difference in AC or parietal cortex neuron sensitivity is observed when switching from passive listening to task engagement, even in fully trained animals[39,75,76]. Therefore, the variables associated with task engagement during practice may temporarily dominate the previously learned signal. Further analysis of the geometric measures of dimensionality and radius of manifolds, as well as the FR profile and coding sparsity of neurons within each manifold, should offer a better understanding of such network reorganization.

Given the dense, reciprocal connections between AC and other sensory and association cortices (e.g., visual, parietal, frontal cortices, striatum, and hippocampus[77–81]), we predicted that the AC network would integrate both task-specific auditory cues and relevant social cues during exposure, thereby facilitating subsequent task performance. We found that information related to the behavior of the performing demonstrators was, in fact, represented in the AC of *social observers* (Fig. 5). Specifically, significant changes in *social observer* AC activity during exposure sessions were time-locked to demonstrator

trial initiation and reward periods. The *social observers'* anticipatory response to the demonstrator's nose-poke during late exposure may be linked to the activity of neuromodulatory signals. For example, dopaminergic signaling has been implicated in auditory social learning[46,82]. In principle, dopaminergic or other neuromodulatory signals could be recruited by demonstrator-associated sounds such as vocalizations (60% of demonstrators' vocalizations were found to be initiated around the time of trial initiation[46]; see Fig. 4), or movement-generated sounds, or the sound of demonstrator chewing. Since the anticipatory LFP signal during practice is correlated with the *social observer's* subsequent behavioral performance, it is likely to reflect a modulatory signal linked to increased attention (nucleus basalis), reward prediction (VTA), or arousal (locus coeruleus) occurring both during the late exposure and practice sessions.

Overall, these results suggest that the encoding of both social and sensory information during exposure may later enhance task acquisition during practice. This finding is in line with previous studies showing that learning to discriminate between songs is enhanced by social exposure to a demonstrator that is performing the task[83]. Similarly, while juvenile songbirds can learn species-specific vocalizations from playback alone, learning is largely facilitated by social exposure to a tutor[84–86]. In human infants, the ability to discriminate between foreign speech sounds is lost by one year, while exposure to a live foreign speaker, but not an audiovisual recording, restores this ability[87]. While there may be differences between social learning of social tasks (e.g., vocalization or speech acquisition) as compared to non-social tasks (as examined in the current study), we cannot exclude the possibility that an overlapping set of neural mechanisms contribute to the acquisition of both social and non-social tasks.

It is still unclear which signals from the demonstrator potentiate the magnitude of plasticity observed in the *social observer's* AC. One possible framework is that social learning is induced by vicarious reinforcement[88]. In fact, there is evidence that implicit reward is accompanied by dopaminergic signaling[89–94]. Similarly, vocal learning in songbirds is associated with dopamine release in the cortical song nucleus HVC, and song learning was completely blocked when dopamine fibers were eliminated prior to exposure with a tutor[82]. Furthermore, exposure to a live singing tutor selectively activates dopamine neurons in a pupil bird, while playback of the song of an adult finch from a speaker fails to evoke dopaminergic activity[82]. We have also found that auditory social learning in gerbils can be suppressed with a D1/D5 dopamine receptor antagonist and facilitated with a D1/D5 dopamine receptor agonist[46]. This mechanistic framework will likely expand to include regions downstream of AC that process social signals, such as vocalizations[27,77,95–97], and other neuromodulatory regions implicated in social learning[93,94,98,99].

## Methods

### Experimental animals
Gerbil (*Meriones unguiculatus*, n = 41) pups were weaned at postnatal day (P) 30 from commercial breeding pairs (Charles River). Littermates were caged together but separated by sex and maintained in a 12 h light/dark cycle. Animals for each of the different experimental protocols came from at least 3 different litters.

### Ethics statement
All procedures related to the maintenance and use of animals were approved by the Institutional Animal Care and Use Committee at New York University, and all experiments were performed in accordance with the relevant guidelines and regulations.

### Behavioral setup
As described previously[36,46], gerbils were placed in a plastic test cage (dimensions: 0.4 × 0.4 × 0.4 m) that was housed in a sound attenuation booth (Industrial Acoustics; internal dimensions: 2.2 × 2 × 2 m) and observed via a closed-circuit monitor. Auditory stimuli were delivered from a calibrated free-field tweeter (DX25TG0504; Vifa) positioned 1 m above the test cage. Sound calibration measurements were made with a ¼ inch free-field condenser recording microphone (Bruel & Kjaer). A pellet dispenser (Med Associates Inc., 20 mg) was connected to a food tray placed within the test cage, and a nose port was placed on the opposite side. The nose port and food tray were equipped with IR emitters and sensors (Digi-Key Electronics; Emitter: 940 nm, 1.2 V, 50 mA; Sensor: Photodiode 935 nm 5 nS). Stimuli, food reward delivery, and behavioral data acquisition were controlled by a personal computer through custom MATLAB scripts and an RZ6 multifunction processor (Tucker–Davis Technologies).

### Stimuli
As described previously[36,46], the Go stimulus consisted of amplitude-modulated (AM) frozen broadband noise tokens (25 dB roll-off at 3.5 kHz and 20 kHz) with a modulation rate of 12 Hz and a modulation depth of 100%. The Nogo stimulus was similar to the Go stimulus except for the modulation rate, which was 4 Hz. Both Go and Nogo stimuli had a 200 ms onset ramp, followed by an unmodulated period of 200 ms, which then transitioned to an AM stimuli. The sound level used was 55 dB SPL.

### Demonstrator training procedure
Demonstrator gerbils were trained by the experimenters on a sound discrimination task (n = 18, 8 females, age = 120.3 ± 31.6). The demonstrators were placed on controlled food access prior to the start of training, and all animals were trained using an appetitive reinforcement operant conditioning procedure[36,46]. Animals first learned to approach the food tray and receive food pellets (Bio-Serv) when the Go stimulus (12 Hz AM noise) was played. Animals were then trained to reliably initiate Go trials independently by placing their nose in the port. Once animals were performing a minimum of 80 Go trials with a hit rate >80%, Nogo trials were introduced. The probability of Nogo trials was kept at 30% in order to keep the animal motivated to perform the task. Nogo trials were paired with a 4-s time-out during which the house lights were extinguished, and the animal could not initiate a new trial. The presentation of the Go and Nogo trials was randomized to prevent animals from developing a predictive strategy.

Responses were scored as a Hit when animals approached the food tray to obtain a food reward upon Go trials. If animals re-poked or did not respond during the 5-s time window following a Go stimulus, it was scored a Miss. During Nogo trials, responses were scored as a False Alarm when animals incorrectly approached the food tray. If animals re-poked or did not respond during the 5-second time window following a Nogo stimulus, then it was scored a Correct Reject. A performance metric, *d prime* (*d'*), was calculated for each session by performing a z-transform of both Hit rate and False Alarm values: $d' = z(\text{Hit rate}) - z(\text{False Alarm rate})$[100]. To qualify as a demonstrator, animals were required to perform the task with a *d'* > 1.5.

### Social exposure sessions
During each social exposure session, a demonstrator gerbil performed the discrimination task in the presence of a naïve same-sex *social observer* (Fig. 2a, left; n = 18, 8 females, age = 119.5 ± 25.1). No animals were excluded from the study. All demonstrators and *social observers* were not littermates nor cage-mates. Social learning with a cage-mate demonstrator versus a non-cage-mate demonstrator was assessed previously, and no significant difference was found[36] (see Supplementary Fig. 3). Both the demonstrator and the *social observer* were placed on controlled food access. An opaque divider (acrylic sheet) was placed within the test cage to separate the observer from the demonstrator[36]. The opaque divider blocked access to visual cues during the exposure sessions; however, the *social observers* maintained access to auditory, tactile, and olfactory cues. The *social*

*observer*s were also exposed to the Go and Nogo sound stimuli as the speaker was located 1 m above the test cage. A nose port and food tray were present only in the demonstrator's compartment, allowing the demonstrator to initiate and perform trials. In each exposure session, the *social observers* were exposed to a minimum of 80 Go trials and 20 Nogo trials performed by the demonstrator gerbils. After five consecutive daily exposure sessions, the divider was removed after the final day of exposure, and the *social observer* was allowed to practice the task (Fig. 2a, right).

### Non-social exposure sessions

During each non-social exposure session, naïve *non-social exposure* animals were separated from an unoccupied demonstrator compartment by an opaque divider (Fig. 2b, left). The experimenter triggered Go and Nogo stimuli from outside the test cage, as well as time-outs during which the light was extinguished inside the sound attenuation booth to simulate False Alarms and delivery of pellets into the food tray to simulate Hits. Misses following Go trials and Correct Rejects following Nogo trials were also triggered. Thus, during the exposure sessions, the *non-social exposure* gerbils ($n = 5$, 2 females, age = 122.6 ± 19.2) retained access to the Go and Nogo sound stimuli and all task contingencies in a non-social manner. In each exposure session for the *non-social exposure* gerbils, the experimenter triggered the same number of Go/Nogo trials and the same responses (i.e., hits, misses, correct rejects, and false alarms) as performed by demonstrator gerbils in the Social exposure paradigm. After five consecutive daily exposure sessions, the *non-social exposure* gerbil was permitted to perform the task (Fig. 2b, right), similarly to the *social observers*.

### Practice sessions

During the first and second practice sessions, both the *social observers* and *non-social exposure* animals were given the benefit of no more than five experimenter-triggered Go trials. These experimenter-triggered Go trials were initiated only when an animal was touching the nose port. This method of manually initiating Go trials was identical to the one used to train demonstrators in order to maximize the animal's interest in the nose port object. Except for these experimenter-triggered Go trials, all Go trials were initiated by the *social observers* or the *non-social exposure* animals. Once an animal was reliably initiating Go trials and performed >25 Hits, Nogo trials were introduced. False Alarm trials were paired with a 2-s time-out on the second day of the Nogo trial introduction. For all following practice days, a 4-s time-out was used when animals were False Alarmed. A $d'$ was computed for all practice sessions during which a minimum number of 15 Nogo trials were presented. To limit the contributions of olfactory and gustatory cues around the nose port and food tray for the *social observers* and the *non-social exposure* animals during the first day of practice, the cage, including the nose port and food tray, was cleaned with alcohol wipes prior to the first practice session. All practice sessions lasted a minimum of 20 min.

### Cannula implants

Gerbils were anesthetized (isoflurane 2%), placed in a stereotaxic frame, and an incision was made along the midline. The skin and fascia were removed to expose the skull. Bone screws were inserted into both frontal bones, and craniotomies were made dorsal and medial to each AC. A double guide cannula (26 gauge, 3 mm cannula length, 1.2 mms center-to-center distance; C235GS-5-1.2/SPC; Plastics One) was angled 20° in the mediolateral plane. The rostral-most cannula in each hemisphere was positioned 3.9 mm rostral and 4.8 mm lateral to lambda. The cannulae were secured to the skull with dental acrylic. Dummy cannulae (33 gauge, 3.2 mm cannula length, C235DCS-5/SP; Plastics One) were inserted to keep the guides clear and were secured in place with a brass dust cap (303DC/1B; Plastics One). Animals were allowed to recover for at least 1 week before being placed on

controlled food access and the start of the behavioral paradigm. At the termination of each experiment, histology confirmed that infusions were centered within the AC without damage to the AC itself (Supplementary Fig. 2).

### Cannula infusions

Prior to each exposure session, gerbils were briefly anesthetized (2% isoflurane). Infusion cannulae (33 gauge, 4 mm cannula length, C235IS-5/SP; Plastics One) were inserted and connected to PE-50 tubing for the infusions of either saline or muscimol. The concentration of muscimol (Abcam) used was 1 mg/ml (dose infused: 0.5 μL per hemisphere at a rate of 0.1 μL/min[39]). For control infusions, physiological saline solution (0.9% NaCl) was used (dose infused: 0.5 μL per hemisphere at a rate of 0.1 μL/min). Animals were allowed to recover in a recovery cage for 15–20 min before each exposure session.

Prior to starting the social exposure session with a demonstrator, all infused animals were monitored for a few minutes in the test cage to assess their locomotion and general behavior. No animals required a longer recovery period, and all animals were alert and engaged and displayed proper posture and normal motor functions.

### Electrode implants

Surgical procedures for electrode implantation were similar to those for cannulae implantation, described above. Gerbils were anesthetized (isoflurane 2%), placed in a stereotaxic frame, and after exposing and drying the skull, bone screws were inserted into both frontal bones. A craniotomy was made in the left parietal bone, dorsal and medial to the AC. A wireless silicone probe array with either 16 or 64 recording sites was implanted in the left AC (Neuronexus, model A4×4-4 mm-200-200-1250-H16 and model Buszaki64_5×12-H64LP_30 mm). The probe was affixed to a custom-made manual microdrive which allowed for subsequent advancement of the probe. The probe was inserted at a 25° angle in the mediolateral plane, such that advancement of the probe allowed sampling of multiple sites passing roughly tangentially through a cortical layer. The rostral-most shank of the array was aimed at 3.9 mm rostral and 4.6–4.8 mm lateral to lambda. A ground wire was inserted in the right caudal hemisphere, and the apparatus was secured to the skull via dental acrylic. Animals were allowed to recover for at least 1 week. At the termination of each experiment, the location of recording sites was confirmed to be in AC by histology (Supplementary Fig. 2).

### Histology

Animals were deeply anesthetized with sodium pentobarbital (150 mg/kg), and electrolytic lesions were made through one contact site via passing current (7 mA, 5–10 s). Animals were then perfused with phosphate-buffered saline and 4% paraformaldehyde. Brains were extracted, post-fixed, and sectioned on a vibratome (Leica). Sections were wet mounted onto gelatin-subbed slides using a fluorescent mounting solution containing DAPI (Vector Laboratories) and inspected under an upright microscope (Revolve Echo). Electrode tracks were reconstructed offline and compared to a gerbil brain atlas[101] to verify targeted core AC regions.

### Electrophysiological recordings

Physiological data were acquired telemetrically from freely moving *social observers* and *non-social exposure* animals while they were either in the exposure or practice sessions using a wireless head stage and receiver (W16 or W64, Triangle Biosystems). Analog signals were amplified and digitized at a sampling frequency of 24,414 Hz and transmitted to a digital signal processor (Tucker–Davis Technologies; 16 channel recordings: TB32 to RZ5; 64 channel recordings: PZ5 to RZ2), then sent to a PC for storage and post-processing. Electrophysiological data underwent common average referencing and were bandpass filtered at 300–5000 Hz. Significant noisy portions of the

signal that were induced by extreme head movements were removed by an artifact rejection procedure. Open-source spike sorting packages were used to extract and cluster spike waveforms (16-channel recordings: UltraMegaSort 2000; 64-channel recordings: KiloSort[102]). Manual inspection of spike waveforms was conducted in Phy[103]. Through principal component clustering and manual confirmation, the extracted waveforms were further sorted into clusters classified as single or multi-units. The single unit's quality was verified using several metrics, including clear separation in principal component space from other clusters, clear refractory periods (<10% refractory period violations), and waveform amplitudes above the noise floor throughout the recording session. Units that did not meet these criteria were classified as multi-units. All sorted spiking data were analyzed with custom MATLAB scripts.

We recorded a total of 864 units (396 single units) for the *social observers* and a total of 1019 units (373 single units) for the *non-social exposure* animals:

1. For the early epoch (i.e., exposure sessions 1 and 2), we identified 416 units (179 single units) for the *social observers* and 233 units (107 single units) for the *non-social exposure* animals.
2. For the late epoch (i.e., exposure sessions 4 and 5), we identified 210 units (109 single units) for the *social observers* and 237 units (97 single units) for the *non-social exposure* animals.
3. For the practice epoch (i.e., all practice sessions), we identified 238 units (108 single units) for the *social observers* and 549 units (169 single units) for the *non-social exposure* animals.

For Fig. 3b–d, both single- and multi-units were included in the analysis. For all subsequent figures and analyses, only single units were used.

Figure 3f only included single units identified. Cells were labeled as regular spiking or narrow spiking according to their spike width (negative trough to first positive peak; boundary placed at 0.43 ms[104]). We observed a similar proportion of narrow-spiking cells for both groups: *social observers* (early, 24%; late, 23%) and *non-social exposure* animals (early, 26%; late, 25%).

Figure 3g included a subset of single unit data recorded from the same animal across exposure days, from the same general recording area (no advancement in probe across exposure days), from the same specific electrode site (63 recording sites), with similar spike waveform (visual comparison), and with similar spontaneous FR (±2 Hz; mean = 0.77 ± 0.89 Hz).

## Neural response properties
Each unit's driven FR was calculated across a time period of spike trains corresponding to the initial onset of the modulation of the stimulus. Spontaneous FR (Hz) was calculated across 200 ms prior to the nose poke. The strength of stimulus synchrony for each unit across tested AM rates and depths was represented by VS[105]. The VS could range from 0 (no synchrony) to 1 (all spikes at identical phase). The statistical significance of the VS was evaluated by the Rayleigh test of uniformity[106] at the level of $p < 0.001$.

## Neurometric classifier
We adopted a pattern classifier analysis to further assess the cortical encoding of AM for individual AC units[54]. The spike pattern metric utilized Euclidean distance to quantify the dissimilarity between two spike trains in response to each AM stimulus (1000 ms) in high-dimensional space[107], using a leave-one-out template-matching procedure. The spike pattern classifier was implemented as follows: First, for each individual unit, test trials consisted of one randomly selected spike train (bin size = 10 ms) from a Go trial and one randomly selected spike train from a Nogo trial. Each Go and Nogo template was composed of all other trials other than the test trials. The test trial was assigned to the Go or Nogo template based on the smallest difference in Euclidean distance between the test and the mean of template trials. Test and template trials were selected randomly, and spike train classification was repeated 250 times to minimize selection biases. Classification of trial to template assignments was scored as follows: Go test trials were labeled as Hits or Misses if they were assigned to the Go or Nogo template, respectively. Likewise, Nogo test trials were labeled False Alarms or Correct Rejections if they were assigned to the Go or Nogo template, respectively (Fig. 3e). The percentage of Hit and False Alarm scores was calculated across repetitions and z-scored to obtain a neural classifier-based $d'$ value. Animals with too few single units during either the early or the late exposure epoch were excluded from the analysis (<10 single units; 1 *non-social exposure* animal was excluded). Separate analyses of single versus multi-unit populations revealed no systematic differences. Comparison between changes in neural $d'$ obtained by the spike pattern classifier and changes in spontaneous FR from exposure to practice ($r = 0.26$, $p < 0.0001$) suggests that FR contributes to information used by the spike pattern classifier analysis in computing neural $d'$.

## Population neural $d'$
We used a previously employed linear classifier readout procedure[47] to assess AM rate discriminability across a population of AC single units. Specifically, a linear classifier was trained to decode responses from a proportion of trials to each stimulus set (Go vs. Nogo). Spike count responses from N neurons were counted across 10 ms bins to T trials of S stimuli throughout 1000 ms of the AM stimulus and formed the population "response vector." Since there were far more Go trials than Nogo trials, we randomly subsampled a proportion of Go trials to match the number of Nogo trials (Fig. 4a). An SVM procedure was used to fit a linear hyperplane to 50% of the data set ("training set"). Cross-validated classification performance was assessed on the average vectors (one for Go trials and one for Nogo trials) of the remaining 50% ("testing set"). We repeated the above process for 250 iterations with a new randomly drawn training and testing set for each iteration. Performance metrics included the proportion of correctly classified Go trials (Hits) and misclassified Nogo trials (False Alarms). Next, we converted population decoder performance metrics into neural $d'$ values. The SVM procedure was implemented in MATLAB using the "fitcsvm" and "predict" functions with the "KernelFunction" set to "linear." This analysis was conducted for sessions with ≥5 simultaneously recorded units. Hence, animals with too few simultaneously recorded single units in every single session were excluded from the analysis (<5 single units; 2 *non-social exposure* animals and 1 *social observer* were excluded).

## Geometric analysis
We used the mean-field theoretic manifold analysis technique[48–50] to study the geometric properties of the stimulus manifolds, including their manifold capacity, radius, and dimensionality. Similarly to the analysis of population neural $d'$, we used the spike count responses across 10 ms bins to T trials of S stimuli throughout 1000 ms of the AM stimulus. Next, we defined 2 object manifolds of neural activations, corresponding each to the Go (12-Hz) and the Nogo (4-Hz) stimuli. We calculated the manifold capacity by averaging 30 repetitions of the following subsampling process. For each session of an animal, we randomly subsampled half of the Nogo trials and the same amount of Go trials (Fig. 4b–d). Together, the neural activations corresponding to these trials form two subsampled manifolds. Then, we ran the mean-field theoretic manifold analysis to calculate the manifold capacity of these subsampled manifolds. Finally, we used the average of these 30 repetitions as the manifold capacity for each given session. In line with the population neural $d'$, animals with too few simultaneously recorded single units in every single session were excluded from the analysis (<5 single units; 2 *non-social exposure* animals and 1 *social observer* were excluded).

Briefly, the manifold capacity, $\alpha$, captures the geometric properties of each object manifold and refers to the maximum number of object manifolds that can be linearly separable per neuron in a given population of neural activities. The coding efficiency of object manifolds can be captured by the ratio $\alpha = P/N$ where $N$ is the number of neurons, and $P$ is the number of object manifolds with random locations in the N-dimensional neural response space. When $\alpha$ is small, there are few manifolds in the high dimensional space, where it is easy to find separating hyperplanes. As more manifolds are added to the high dimensional space, $\alpha$ increases, and it becomes harder to find separating hyperplanes. The critical manifold capacity, as computed in our analysis, refers to the maximum number of object manifolds $P$ that can be linearly separated given N neurons with high probability when binary labels are randomly assigned to each manifold. Here, the intuition is that when critical $\alpha$ is small, the geometry of manifolds is such that you can linearly classify a few manifolds in the high dimensional space (tangled manifold representations, usually high dimensional or large in size). If critical $\alpha$ is large, then the geometry of manifolds is such that you can classify many such manifolds (untangled manifold representations, usually low dimensional or small in size). This quantity can be estimated from the statistics of anchor points and representative support vectors defining the optimal separating hyperplane[48]. In fact, the largest possible value of capacity is 2 and is achieved when the manifold vanishes to a single point[48,108]. To study the underlying driving factors of a change in capacity, we also computed two additional quantities from the manifold analysis: the manifold anchor radius, $R$, and the manifold anchor dimensionality, $D$. $R$ and $D$ capture, respectively, the effective radius and dimensionality of the manifolds during linear classifications. The mean-field theoretic manifold theory shows that the capacity, $\alpha$, is inversely related to both $R$ and $D$[48].

## Local field potentials

Raw electrophysiological recordings were downsampled to 1 kHz from a subset of five randomly chosen active channels to calculate LFPs. For each session, we computed the LFPs by averaging the raw data time-locked to trial onset. The LFPs were baseline-corrected by subtracting the average value during all no-sound periods, then averaged by trial type (Go and Nogo trials), as well as response types (Hit, Miss, False Alarm, and Correct Rejection). We examined the frequency decomposition of the LFPs by using an FFT and limiting the analysis to the AM response. The mean Fourier amplitude spectrum for each trial type was also normalized in amplitude to compare the FFT across sessions. The amplitude of the peak corresponding to the modulation frequency of the stimulus (4 or 12 Hz) was extracted for further statistical analysis using a mixed model ANOVA.

Next, we calculated the Morlet Wavelet transformed time-locked to trial onsets, and averaged by trial type (Go and Nogo trials), as well as response types (Hit, Miss, False Alarm, and Correct Rejection). As for the LFPs, we applied a baseline normalization by computing the averaged spectrum during all no-sound periods and subtracted this spectrum from the scalogram of each individual trial. Thus, the scalograms do not represent absolute power but power variations relative to the mean power during the no-sound epochs. This process also suppresses the 60 Hz electrical hum. For each trial, the scalogram was computed from −1 second to 3 second with respect to the sound onset (0 s), with a further 226-ms margin to avoid edge effects in the wavelet representation. The wavelet analysis was limited to 6 Hz (minimum frequency) to limit the analysis time window so as to not overlap with the previous or subsequent trial. We next computed the average power in a low-frequency band (6–20 Hz) and a high-frequency band (20–80 Hz) and compared the average power to the baseline (zero) in defined regions of interest for statistical analysis (Fig. 5).

A finer decomposition of frequency bands into 5 distinct bands (6–8 Hz, 6–12 Hz, 12–20 Hz, 20–40 Hz, 40–80 Hz) revealed grossly similar results in the first three bands and the last two bands. We thus chose to merge the first three bands and the last two bands. Note that for practice sessions analysis, we limited our analysis to practice sessions with behavioral $d' > 1.5$.

## Performance measures and statistical analyses

A performance measure ($d'$) was calculated for each animal: $d' = z$(Hit rate) − $z$(False Alarm rate). Hit and False Alarm rates were constrained to floor (0.05) and ceiling (0.95) values. A $d'$ was computed for every session during which the animal performed >15 Nogo trials. For the computation of the mean $d'$ line, we used all values of $d'$ and attributed a zero to all NaN values of $d'$ (i.e., when an animal was initiating <15 Nogo trials, no $d'$ value could be computed). For all statistical tests regarding the mean number of days taken by each observer group to reach a criterion $d'$ of 1.5, only actual $d'$ values were used. All group-level statistical tests and effect size calculations were performed using JMP Pro 14.0 on a Mac platform or custom-written MATLAB scripts (MathWorks) that incorporated the MATLAB Statistics Toolbox. The Shapiro Wilk test of normality was performed prior to parametric tests, such as ANOVA. Non-normally distributed data was examined using non-parametric tests, such as the Wilcoxon rank sum test and Steel–Dwass non-parametric comparisons (two-sided). For all post hoc multiple comparisons analyses, alpha values were corrected, as indicated. For more details, see Supplementary Note 1.

## Reporting summary

Further information on research design is available in the Nature Portfolio Reporting Summary linked to this article.

## Data availability

The data generated in this study have been deposited on an NYU repository (NYU app box) and can be accessed here: https://nyu.box.com/s/rv9m4ionulfk42nkbamff1h706xaeko7. Raw electrophysiological data can be shared upon request to the first author (np64@nyu.edu; ongoing further computational analysis by co-authors). Source data are provided in this paper as a single Source Data file.

## Code availability

Custom scripts for data analysis have been deposited on an NYU repository (NYU app box) and can be accessed here: https://nyu.app.box.com/folder/190089728369?s=rv9m4ionulfk42nkbamff1h706xaeko7 in the folder: Matlab code.

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

## Acknowledgements

The work was supported by the Fyssen Foundation (N.P.) and R01DC020279 (D.H.S). The authors thank J.A. Charbonneau for his contribution to pilot data collection.

## Author contributions

N.P. and D.H.S. designed research and secured funding; N.P. performed all behavioral experiments and collected all in vivo physiological data; N.P. analyzed the data with J.Y., L.V., C.C., and S.C.; N.P. and D.H.S. wrote the paper.

## Competing interests

The authors declare no competing interests.
