## [Peer Review File · Nature Communications]

REVIEWER COMMENTS

Reviewer #1 (Remarks to the Author):

The manuscript by Paraouty et al. is a beautiful study of social learning, as it applies to learning an auditory-based operant conditioning task. The authors use a paradigm they have previously published to show that the auditory cortex in gerbils is necessary for auditory social learning. Furthermore, they record neuronal activity during the social learning and non-social exposure control task and find differences in how the auditory cortex encodes conditioned auditory stimuli. These differences correlate with the learning rate. The authors also show exciting preliminary data indicating that social cues might be encoded in the auditory cortex and might impact how conditioned auditory stimuli are represented. The paper is well written, and the results are meaningful and important.

I have several comments and concerns which I will list below in no particular order:

1. Figure 1, given that anesthesia was required for infusion, do we know that the animals had enough time to recover? Were their reflexes checked or some other measures of full recovery after anesthesia?
2. Figure 1 – it is possible that repeated muscimol inactivation of AC might affect the proper function of AC, and could explain the subsequent changes in learning curve (compared to saline controls). I think a good control would be to perform the same inactivation experiment during the non-social exposure and see if it affects the learning curve during the non-social procedure.
3. Figure 1 – the first practice session comes immediately after the 5th (last) exposure to the demonstrator. During the inactivation experiment, is it possible that the auditory cortex is still inactivated during the first practice session and the animals have a false start that prevents them from initiating the learning trajectory right away? Could the authors exclude this possibility?
4. A general question regarding the behavioral approach – given that the animals use the same nose port and food tray as the demonstrator, could the gerbils use olfactory and gustatory social cues during practice to faster acquire the task?
5. For Figures 1b and 2c – it is odd to include the brain diagram within the data plot (as if they too were measurable). Please move the y-axis next to the data
6. Why was there a cutoff of 15 Nogo trials for measuring d' ?
7. Figure 3 – do the authors know that they have a similar proportion of excitatory and inhibitory neurons (even if only 'putative') in the neuronal population recorded in social vs non-social exposures? Could a slight difference in proportion explain the (not as dramatic) differences in neuronal responses?
8. Figure 3f – it appears that the neural d' for early non-social exposure is higher than the d' for early social exposure. I don't think the authors commented on a potential cause. But it would explain why the difference from early to late exposure is higher in social than in non-social.

9. Figure 3f – in the text (lines 258-263), the authors describe these data in terms of median d' . But this plot shows something else – the Neural d' for the bottom 80% of neurons. It is unclear why the authors don't describe the stats for this representation or why they don't plot the median d' .
10. Figure 3f – could not see a color legend, but assumed it's the same as in 3b,c,d
11. Figure 3g – are the neurons colored in blue and orange experiencing a significant increase in d' ? I did not see that specified, meaning no test for significance was mentioned (maybe I missed it?).
12. Figure 3g – please use Fisher's exact test to see if proportions are significantly different.
13. What might be the social cues that induce changes in LFP? Have the authors tried to record vocalizations?
14. The LFP data are a little underwhelming. Have the authors tested to see if the LFP signal correlates with behavioral performance?

Reviewer #2 (Remarks to the Author):

In this study, Parouty et al. investigated the role of auditory cortex (AC) plasticity for task acquisition during social learning. To this end, the authors utilized a social learning paradigm previously established in the same lab that consists of a demonstrator and an observer animal and record from or transiently inactivate AC. The authors provide convincing evidence that AC plasticity in observers improves neural sensitivity to task relevant metrics, which is correlated with social exposure to the demonstrator (i.e., improvements are not present after passive cue exposure). They further report that this AC plasticity supports subsequent behavioral performance / task acquisition. In line with these results, inactivation of AC of observers delayed task acquisition. Finally, LFP analyses are presented demonstrating that changes in the dominant frequency bands in AC of experienced observers are induced by social cues. Together, the authors thus provide novel insight how sensory information during social learning alters neural coding in AC and how this plasticity helps subsequent learning of the observed task.

The results are presented in a highly comprehensible and elegant fashion that make the paper a joy to read. The findings are explained very clearly and discussed coherently in context of the current literature of social learning and cortical plasticity, and the conclusions are consequently well supported and convincing. The work will be of great interest to neuroscientists interested in sensory processing and (social) learning and will represent an important reference for future studies in this field.

I only have a few comments for clarification purposes.

Line 184: Can you really claim "causality" in terms of social learning at this stage? One might argue that the transient inactivation is somewhat too unspecific, as it could also suppress any processing and

plasticity not directly related to social learning, and might rather induce general learning or acquisition deficits of some sorts. To claim specific causality, I would recommend testing AC inactivation in non-social exposed animals to control if no delay in task acquisition occurs compared to a saline-infused non-social exposed group.

Line 246: Please give exact p values (instead of “ $p > 0.05$ ”). Variability is quite high for the practice groups, and it would be interesting to know if the p values indicate a trend, even if not reaching the selected alpha criterion.

Line 266: How did you assess the identity of neurons across sessions? I.e., how did you make sure it is (or is not) the same unit across days? I could only find information about the general sorting procedure in the Methods.

Line 396: I agree that this represents a major finding. However, it should be noted that significant plasticity (in terms of altered rate, CoeffV, and VS) also occurred prior to practice in the non-social exposure group (Fig 3.b). Panels 3f and 3g indicate that these changes might be less “advantageous” for decoding, but nonetheless they suggest some sort of improvement.

Figure 3: It took me some time to figure out the color code (shades of blue and red), since the respective labels are positioned at the bottom of panel b of a very busy figure. Maybe one can think about another / additional way to convey the color code?

Figure 4: It somewhat struck me that all metrics shown in panes a – d exhibit a rather large difference between the last day of exposure and start of practice for the social observers, but not for the non-social group. How do the authors interpret / explain this social-specific discrepancy? I would have expected a continuation of each metrics’ values from exposure to practice, not the least because the authors made the argument that the change in the metrics during exposure might be beneficial for task acquisition.

Reviewer #3 (Remarks to the Author):

The manuscript by Parouty et al. entitled ‘Sensory cortex plasticity 1 supports auditory social learning’, explores the role of the auditory cortex (AC) in social learning of an auditory discrimination, as well as the changes in AC neuronal activity associated with this learning. The authors use a social learning paradigm previously established by the authors (Parouty et al., 2020 and 2021), where they demonstrated that gerbils exposed to the sounds and smells of a demonstrator performing an auditory go/no-go discrimination task, subsequently learned the task faster than gerbils that had only been exposed to subsets of the task parameters not including the demonstrator. Here the authors exposed two groups of gerbils. The social observer group underwent social exposure, while the non-social exposure group was exposed for the same amount of time to the sounds of the experimenter performing the task. All gerbils were then trained in the task. The social observers learnt the task faster than the non-socially exposed gerbils. The authors tested the hypotheses that the auditory cortex plays a role in social learning of an auditory discrimination and that social learning enhances neuronal sensitivity to task-related sounds, finding evidence for both. Muscimol inactivation of the AC during

social exposure led to a decrease in the exposure-induced speed of subsequent learning indicating that the AC contributed to this facilitation. Recordings from the AC during social and non-social exposure and subsequent training, revealed that while both social and non-social exposure led to improved temporal precision of sound representation, only social exposure led to an increase in neuronal discrimination, measured through neuronal d' , and an increase in neuronal classification and representation stability, measured using MFT manifold analysis. The study is a very important and elegant contribution to the understudied mechanisms of social learning. It is well designed, uses sufficient subject numbers and it is generally appropriately controlled.

Some improvements are suggested below:

General comments

The authors use, throughout the introduction and the discussion, examples of vocalization learning. To me and a priori, there is a big difference between social learning of non-social tasks and the learning of social behaviors such as communication. Since there are plenty of example for social learning of non-social tasks and this is what is being tested in this work, I think it would be appropriate to restrict examples to this domain. The potential distinction of lack of it between social learning of social and non-social behaviors could, and should, nonetheless be included in the text.

The comparison with practice performance in the absence of social exposure is missing at points (see comments below) and would be useful. In general, going through the different non-social and social (with yoked reward, for example) exposures used by the authors, one cannot help by wondering whether there is something categorically different about the social aspect. Alternatively, an equally interesting, it could be that exposure learning is cumulative and more efficient the more information is provided, something that is maximized when a social demonstrator is present. I would appreciate a discussion of this point in the paper with evidence for and against.

In relation to the previous point, is the quality of the learning in the social and non-social exposure groups different? It seems that some aspects of the LFP changes that are present during practice in the social group are not present in the non-social exposure group. Some information about the differences, beyond the presence of a gerbil, between the social and non-social exposure patterns would help in this direction.

The analysis of the encoding of social cues is important, although a lack of modulation by specific social cues would not diminish the importance of the findings of increased neuronal sensitivity to task-related sounds under social exposure. The transition to LFP analysis and the choice of frequency bands is a bit sudden. Some guidance through the reasons for this transition would be welcomed.

Statistics in general: I would suggest the use of linear models, or at least take into account repeated measures and multiple comparisons.

Specific comments

Abstract:

Line 59: I am not sure that 'human aureal communication' is the best example given the focus of the manuscript. Human aureal communication is learned through social interaction but it is not a behavior that is learned just by observation. Efficient learning in infants depends on the communication being directed at them. While it undoubtedly has components of social learning, it might be a special case because it is, itself, a social behavior. The number of youtube videos demonstrating all sorts of behaviors, from cooking and knitting to putting together an engine or fixing a watch, demonstrates the extensive use of social learning.

Line 63: for the naïve reader this sentence would be clearer if it were to declare when the manipulation was done and when the effect tested.

Line 67 and 70: I am not sure what 'improvement' (line 67) and 'poorer' (line 70) means. What is the comparison?

Graphical abstract: bottom right graph, the y axis legend 'practice acquisition rate' is misleading. Higher rate of acquisition typically means faster learning but I assume here the higher numbers reflect not higher rate of acquisition but longer time/more trials to reach criterion? The y axis is correct in the 'real' graph.

Introduction:

Second paragraph: As a reader, in order to assess whether the question of the manuscript aims to add one more example of neuronal plasticity to the list or whether it goes a step further in implicating a specific structure and a specific mechanism in a specific social learning behavior, it would be useful to have more detail on the effects that the listed manuscripts found. What does it mean that a structure was 'implicated', 'involved', or 'contributes'?

Line 121: 'Therefore, social experience may have a general effect on sensory encoding of environmental cues, and this initial plasticity may facilitate the acquisition of new behaviors.' I think the concept exposed here is the crux of the matter and could be explained a bit more extensively. What is it meant by 'a general effect on sensory encoding environmental cues'?

Line 133: You could include de Hoz et al., Cerebral Cortex 2018 PMID: 28334281, where early gene expression specifically in AC was shown to be important in discrimination learning.

Results:

Role of AC – muscimol:

Exposure under muscimol: was there any behavioral difference between saline- and muscimol-injected animals during the exposure phase?

According to the methods, day 5 of exposure was directly followed with day 1 of practice. Were the muscimol animals not under the effect of muscimol on day 1 of practice? Please explain and justify the design if this is the case.

The time course of the effect of muscimol is difficult to control. How was it assessed? Do the authors have any evidence that the AC of muscimol-treated gerbils was fully functional during the first 2-3 days of training?

Supplementary figure 1: The delay in learning is difficult to interpret without knowing how a naïve unexposed animal would behave. Based on Parouty et al., 2020 Figure 2c, it looks like the muscimol animals perform similarly, maybe slightly better, than non-socially exposed animals. I think this is an important comparison that should be included in the current manuscript. It is important because it helps interpret the data and, although it is no proof that muscimol has not necessarily left post-exposure long lasting effects that have delayed the learning, it is consistent with t. At the same time, it looks like the delay in learning stems from a delay in making enough go trials. Once no-go trials are introduced, on day 3 for the saline groups and on day 6 for the muscimol group, the rate of learning is faster for the saline group than the muscimol group. Both groups, however, are faster than the untreated and unexposed animals in Parouty et al., 2020 Figure 2c, which need 7 days to reach a d' of 1.5. What do the authors think is the meaning of this?

Figure 1c and supplementary figure 4: It looks like a white rectangle was put over the figure to cover the lower values. It is very disconcerting. I think the figures should be replotted.

Neuron response properties:

Line 232: 'In contrast, no changes in firing rate occurred during silence or during the unmodulated portion of the sound stimulus (see Supplementary Figure 5, $p > 0.05$).' because all other comparisons are including both exposure phases and the practice phase, this sentence leads to believe that there is no difference between all 3 phases, but the firing rate is significantly higher during the practice phase according to Sup. Fig. 5. Please expand.

The ranges of firing rates increase with exposure and practice. Is there a correlation between the firing rate and the neuronal d' ? and do the neurons that improve d' over time also increase their firing rate?

Neuronal sensitivity:

One problem I see with the neural d' is that it doesn't consider the quality of the classification. For example, a spike train during a go trial might be classified correctly as a hit because it is less distant from the go template than from the no-go template and yet be quite different from both. Please include some quantification of distribution of Euclidean distances and, if applicable, describe the distance threshold for inclusion of spike trains.

Paragraph starting on line 265: it is notoriously difficult to demonstrate that the same single unit is being recorded from one day to the next. If the authors are arguing that they recorded from 91 and 86 single units in the social and non-social exposure groups, respectively, across the early and late phases, it would be important to show some quantification of the stability of the spike shapes and the tuning, as well as some example spike shape means across channels for both the early and late phase.

Figure 3h and paragraph starting line 271. The plot is beautiful, yet it could be that the two groups behave differently. While it looks like there is a clear interaction between neuronal sensitivity and behavioral performance, it also seems that the correlation that is clearly observed within the social-exposure group might not be as strong in the non-social exposure group, where the neuronal sensitivity is relatively homogeneous across behavioral performance. Please discuss. Also, please explain the x axis better. I assume it is plotting, for each gerbil, the median, across all units recorded, of the neural d' difference between early vs late.

Line 292; please include an intuition in the text of the MFT manifold analysis, including the capacity and, later (line 306), the radius. This is partly done in the discussion but should be included and expanded in the results.

Paragraph starting Line 314. Please refer to the subplots in Sup. Fig. 6

Social cues:

Line 332: the transition from spikes to LFP is a bit sudden and would benefit from a short justification. Same for the use of the two frequency bands.

Regarding the frequency bands, theta (somewhere between 6 and 12Hz, depending on literature) is associated with locomotion. I am not sure to what extent one would detect theta in AC, but it wouldn't be surprising given its proximity with hippocampus and the fact your electrodes are quite deep in AC. During practice one would expect an increase in theta in the approach to the reward. Would a further dissociation of the low frequency band answer whether the anticipatory increase is locomotion related?

How do the exposed social observers and the non-social exposure animals behave during exposure phase? Do they follow the demonstrator/ experimenter?

Figure 5: the colours are very pastel and the contrast, difficult. I suggest making the poke-bar more distinctive, so that the timing of events in the power trace relative to the poke is easier to see. I also suggest making power traces darker and thicker and to use as much of the width as is possible. The schematic of the sound could represent a lower AM... etc, etc... Just to give you an idea, I have the pdf at 150%, glasses on, and I am still struggling.

Why is the anticipatory increase before the poke not observed in the non-social exposure group during practice?

Please make it explicit in text and/or figure legend that the practice plot only includes sessions with d' above 1.5. It is an important piece of information, but I had to look for the one sentence in the methods.

Sup. Fig. 7a-b, the y axis values are cut or wrong. The plots are very small! Have pity.

Sup. Fig. 7c, please add the color legend to the plot.

What is the peak at about 8Hz that is observed in most traces in both groups and both trial types?

Discussion

Line 438: Please specify the connections between AC and other structures and specifically name those that are speculated to subserve social learning.

Line 442: 'Specifically, social information linked to demonstrator trial initiation and reward elicited significant changes in social observers AC activity.' While likely, strictly speaking there is no evidence that what triggers changes in activity around trial initiation and reward is social information. There might be other sounds that associated with the demonstrator's presence that, if replicated by the experimenter would lead to the same effect. Please rephrase here and other instances. See general comments.

Line 453 and related to previous comment, were vocalizations recorded in this study? If not, since training in previous studies was presumably identical, could the timing of social information such as vocalizations be included here and discussed in the context of LFP patterns?

Methods:

Housing: How were the gerbils housed? If in groups, did they have as partners gerbils from other groups (demonstrator, social exposure etc)?

Non-social exposure method: to better understand whether the difference between this protocol and that for the social exposure results only from differences in the social aspect, it would be useful to have the number of nose-pokes and rewards during both exposure phases, as well as the distribution of time intervals between nose-pokes and rewards.

Response to Referees

Reviewer #1 (Remarks to the Author):

The manuscript by Paraouty et al. is a beautiful study of social learning, as it applies to learning an auditory-based operant conditioning task. The authors use a paradigm they have previously published to show that the auditory cortex in gerbils is necessary for auditory social learning. Furthermore, they record neuronal activity during the social learning and non-social exposure control task and find differences in how the auditory cortex encodes conditioned auditory stimuli. These differences correlate with the learning rate. The authors also show exciting preliminary data indicating that social cues might be encoded in the auditory cortex and might impact how conditioned auditory stimuli are represented. The paper is well written, and the results are meaningful and important.

Specific comments

I have several comments and concerns which I will list below in no particular order:

1. Figure 1, given that anesthesia was required for infusion, do we know that the animals had enough time to recover? Were their reflexes checked or some other measures of full recovery after anesthesia?

For infusions of saline or muscimol, animals were indeed briefly anesthetized (2% isoflurane). All animals were under anesthesia for no longer than 15 minutes in total (around 5 min of infusion per hemisphere).

We have added the following to the Methods Section of the manuscript, Lines 735-741: “All animals were allowed to recover in a recovery cage for 15-20 minutes before each exposure session. Prior to starting the social exposure session with a demonstrator, all infused animals were monitored for a few minutes in the test cage to assess their locomotion and general behavior. No animals required a longer recovery period, and all animals were alert and engaged, and displayed proper posture and normal motor functions.”

In addition, we have clarified in the manuscript that we observed no significant effect related to anesthesia: the saline-infused animals' performance was identical to our previous data on unanesthetized animals from Paraouty et al., 2020. We have added the following to the Results Section Lines 171-175: “No significant difference was found between the number of days taken to reach the criterion d' by the current saline-infused *social observers* and those tested without any infusions or cannulae implants in our previous report (Paraouty et al., 2020, Figure 3; Steel-Dwass nonparametric comparison, $p=0.967$). This suggests that neither the surgery, nor the saline infusions under anesthesia prior to the exposure sessions, impacted the subsequent rate of task acquisition during the practice sessions.”

2. Figure 1 – it is possible that repeated muscimol inactivation of AC might affect the proper function of AC, and could explain the subsequent changes in learning curve (compared to saline controls). I think a good control would be to perform the same inactivation experiment during the non-social exposure and see if it affects the learning curve during the non-social procedure.

As we understand the Reviewer's comment, repeated muscimol exposure could have interfered with AC processing both during the exposure sessions and also produced a longer-term effect that persisted

during the practice sessions. Below, we summarize our previous experience with muscimol in behavioral paradigms, and then provide a new analysis to directly address the Reviewer's concern.

We have previously used bilateral muscimol infusion into AC (Caras and Sanes, 2017). In Caras and Sanes (2017) Figure 4, the authors used an identical dose of muscimol as in the current study (i.e., 1 µg per hemisphere; 1 µL infusion at 0.5 mg/mL) to test the role of AC in perceptual learning (i.e., practice-based improvement). In this experiment, animals received 5 days of muscimol infusion, followed by 5 days of saline infusion. During the subsequent saline infusion days, animals improved their performance at a normal rate as controls. From this, we conclude that muscimol infusion does not have a significant long-term effect on task performance or improvement.

Furthermore, in Caras & Sanes (2017) Supplementary Figure 1, the authors used twice the dose of muscimol as in the current study (i.e., 2 µg per hemisphere; 1 µL infusion at 1 mg/mL). Animals received either this high dose of muscimol or saline on alternate days. The performance during each saline infusion day directly following a muscimol infusion day was identical to control performance (prior to any infusions). These results further confirm the absence of long-term effect of muscimol infusion on AM detection performance.

In the current study, we used an identical dose of muscimol as in Caras and Sanes (2017, Figure 4): 1 µg per hemisphere (note that the volume was halved, and the concentration was doubled, yielding the same dose of muscimol: 0.5 µL at 1 mg/mL), we have assumed that there is also no long-term effect of repeated drug administration, at least not on task learning.

Since the *non-social exposure* animals also learn the task, albeit at a slower rate than the *social observers*, we have concluded that the *non-social exposure* animals do display incidental learning of the task cues. Hence, if we were to perform muscimol infusion on the *non-social exposure* animals, then we would also expect to see a delay in their subsequent learning (i.e., muscimol would block activity evoked by the task cues during the non-social exposure sessions).

Therefore, we have addressed the Reviewer's valid concern with a more nuanced interpretation of our results, Lines 183 - 186: "Although we cannot rule out a long-term effect of muscimol on task acquisition, previous results (Caras and Sanes, 2017) indicate an absence of long-term muscimol effects on psychometric performance. Our results show that the AC may contribute to auditory social learning, and suggests that it may store task-specific acoustic information during the exposure phase."

3. Figure 1 – the first practice session comes immediately after the 5th (last) exposure to the demonstrator. During the inactivation experiment, is it possible that the auditory cortex is still inactivated during the first practice session and the animals have a false start that prevents them from initiating the learning trajectory right away? Could the authors exclude this possibility?

This concern is related to the comment expressed by the Reviewer in point #1 (i.e., the longevity of muscimol's effect on AC). Therefore, our previous answer in #1 should also address this comment.

An additional observation is that, in Caras and Sanes (2017) Supplementary Figures 10 and 11, the authors used identical doses of muscimol as in the current study (i.e., 1 µg per hemisphere) to test performance on an AM detection task. The authors concluded that this dose of muscimol perturbed perceptual learning, but did not grossly perturb psychometric performance in adult gerbils as the

animals displayed excellent detection of AM, and there was no effect on trial completion rates, on false alarm rates nor on reaction times.

Since the behavioral task that we used differs from the Caras and Sanes study, we cannot specifically rule out the possibility that muscimol exposure on day 5 persists into the first practice session and diminishes performance on day 1 of practice.

Therefore, to evaluate this possibility, we have recalculated the difference in rate of task acquisition between the muscimol- and saline-infused animals after excluding the first day(s) of practice:

- after excluding the first day of practice, we still found a significant delay in task acquisition (Steel-Dwass comparison, $p=0.033$), and
- after excluding the first 2 days of practice, we still found a significant delay in task acquisition ($p=0.049$).

This issue is now explicitly acknowledged in the Results section, Lines 188 - 191: “To evaluate whether the effect of muscimol infusion persisted into the first one or two days of practice, we excluded these days from the data analysis and confirmed a significant delay in rate of task acquisition, as compared to the saline-infused *social observers* (after excluding practice day 1, $p=0.033$; after excluding practice days 1 and 2, $p=0.049$).”

4. A general question regarding the behavioral approach – given that the animals use the same nose port and food tray as the demonstrator, could the gerbils use olfactory and gustatory social cues during practice to faster acquire the task?

During practice sessions, olfactory and gustatory cues from the nose port and food tray could contribute to improved task acquisition for both groups of animals: *social observers* and *non-social exposure* animals. Those cues were minimized by regular cleaning of the test cage. Thus, for both groups of animals, the cage was cleaned using alcohol wipes for the nose port and food tray prior to the first day of practice. This is now detailed in the Methods section, Lines 712 - 716 : “To limit the contributions of olfactory and gustatory cues around the nose port and food tray for the *social observers* and the *non-social exposure* animals during the first day of practice, the cage, including the nose port and food tray, were cleaned with alcohol wipes prior to the first practice session.”

5. For Figures 1b and 2c – it is odd to include the brain diagram within the data plot (as if they too were measurable). Please move the y-axis next to the data

We have edited both figures to clarify that no measurements were made from the brain:

- To clarify the brain infusion schematics in Figure 1b, we have added the word “Manipulation” above the infused solution (either “Saline” or “Muscimol”).
- We have removed the brain recording schematics from Figure 2c, and have added the phrase “Animals with chronic electrode implants (Neural data shown in Figures 3-4)” within the blue “Exposure” box.
- For consistency, these changes were also implemented for Supplementary Figures 1, 3, and 4.

6. Why was there a cutoff of 15 Nogo trials for measuring d' ?

To obtain robust measures of d' during practice, we used the same minimum number of Nogo trials (i.e., 15 Nogo trials) for all experiments. The same standard was used in our previous work - Paraouty et al., 2020 and Paraouty et al., 2021.

This measure was taken to avoid erroneous d' measures during early practice days. For example, following 1 Nogo trial and an accidental Correct Reject of the Nogo trial, an animal would end up with an arbitrarily high value of d' suggesting that it is performing the Go-Nogo task well.

7. Figure 3 – do the authors know that they have a similar proportion of excitatory and inhibitory neurons (even if only ‘putative’) in the neuronal population recorded in social vs non-social exposures? Could a slight difference in proportion explain the (not as dramatic) differences in neuronal responses?

In a previous paper from the lab with identical recording electrodes, similar coordinates of electrode implantation and data analysis procedures (see Penikis et al., 2022), the authors found ~25% of putative inhibitory AC neurons (72 narrow spiking cells from 277 single units). In that study, the majority of cells analyzed were regular spiking and hence, putative excitatory neurons (205 cells from 277 single units).

We believe the Reviewer is either referring to Figure 3d or Figure 3f-g.

For Figure 3d - the vector strength, we included both single and multi-units, and as the group difference is minor, we concluded in Lines 260-261: “Together, these results suggest that both social and non-social exposure are associated with changes in AC neuron firing rate and temporal processing.”

For Figure 3f - the neural d' , we included only single-units, making the proportion of putative excitatory and inhibitory neurons relevant to this analysis. Therefore, we have now conducted the same analysis as in Penikis et al., 2022. Cells were labeled as regular spiking or narrow spiking according to their spike width (negative trough to first positive peak; boundary placed at 0.43 ms in line with Penikis et al., 2022). We observed a similar proportion of narrow-spiking cells for both groups: *social observers* (early, 24%; late, 23%) and *non-social exposure* animals (early, 26%; late, 25%). These data are now included in the Methods section, Lines 793 – 796.

Even though the slight differences in proportion (i.e., 1-2%) during early and late exposure may contribute to the minor group differences, we found no significant correlation between spontaneous firing rate and the neural d' computed using the pattern classifier analysis (Early exposure, Pearson’s, $r=0.02$, $p=0.782$; Late exposure, $r=0.07$, $p=0.308$; Practice, $r=0.025$, $p=0.623$). Together, these results indicate that a higher proportion of narrow-spiking cells does not appear to contribute to the findings (i.e., higher d' values).

8. Figure 3f – it appears that the neural d' for early non-social exposure is higher than the d' for early social exposure. I don’t think the authors commented on a potential cause. But it would explain why the difference from early to late exposure is higher in social than in non-social.

The comment refers to the cumulative distributions that were shown in original Figure 3f. It is indeed true that the d' at which each curve crossed 0.8 is higher for the *non-social exposure* animals than for the *social observers*. However, no significant group difference was found when comparing neural d'

during the Early Epoch ($p=0.2938$). Nevertheless, this small difference could in part explain why the *non-social exposure* animals do not show a significant difference from early to late exposure.

While Figure 3f takes into account neural d' of all single units for all animals merged together per group, Figure 3h separates the information of all single units per animal in each group. As is shown in Figure 3h, the median change in neural d' from early to late exposure for all *social observers* is larger as compared to the *non-social exposure* animals. This is true for all animals and overall, a significant difference is observed between the 2 groups (Wilcoxon rank sum test, $X^2(1)=6.56$, $p=0.011$).

9. Figure 3f – in the text (lines 258-263), the authors describe these data in terms of median d' . But this plot shows something else – the Neural d' for the bottom 80% of neurons. It is unclear why the authors don't describe the stats for this representation or why they don't plot the median d' .

The Reviewer is correct. The original version of Figure 3f showed cumulative distributions, yet the manuscript presented median values. To reconcile this difference, we have updated Figure 3f to display box plots with median values, in line with the text in the Results section.

10. Figure 3f – could not see a color legend, but assumed it's the same as in 3b,c,d

The Reviewer is right. We have updated Figure 3f with clear labels.

11. Figure 3g – are the neurons colored in blue and orange experiencing a significant increase in d' ? I did not see that specified, meaning no test for significance was mentioned (maybe I missed it?).

The original version of Figure 3g showed the neural d' for all single units that were tracked from Early to Late exposure. For the line plot, blue or orange represented neurons that displayed an improvement in neural d' , and the bar plots showed only the % of cells that improved versus those that did not change or worsened. No significance analysis was originally performed.

To address the Reviewer's comment, we have updated Figure 3g, and performed statistical analyses. The bar plots now illustrate the % of cells that showed significant improvements, significant worsening, and no significant change. The results indicate the same trend identified previously.

For the associated analysis, we generated 10 unique values of neural d' for each individual cell (as the spike pattern classifier analysis uses one randomly selected spike train from all Go trials as the *Go Test trial*, and one randomly selected spike train from all Nogo trials as the *Nogo Test trial*). Thus, for each individual cell, we computed whether the change in neural d' from Early to Late exposure is significant or not, using two-sample t-tests. This analysis is now included in the Results section, Lines 285 - 293: "First, we generated 10 unique values of neural d' for each individual cell as the spike pattern classifier analysis uses one randomly selected spike train from all Go trials as the Go Test trial, and one randomly selected spike train from all Nogo trials as the Nogo Test trial. Next, we computed two-sample t-tests to examine whether the changes across exposure were significant for each individual AC single unit. Figure 3g shows the mean d' values for each AC single-unit during early and late exposure. For the *social observers*, 62% of single units ($n=56$; Figure 3g, top) displayed a significant improvement in neural d' from early to late exposure. In contrast, for the *non-social exposure* animals, only 22% of single units ($n=19$; Figure 3g, bottom) displayed a significant increase in neural d' ." We have also modified the Figure Legend in line with these new results (Lines 592 - 597).

12. Figure 3g – please use Fisher’s exact test to see if proportions are significantly different.

As suggested by the Reviewer, we have now used contingency analysis to assess the distribution of cells (that significantly improve/worsen; and no significant change) in the 2 groups. A significant group difference was observed in the distribution of cells (Likelihood ratio Chi square, $p=0.0006$). This analysis is now included in the Results section, Lines 293 - 295: “Furthermore, a significant difference was found between the 2 groups of animals in terms of distribution of cells (i.e., significantly improve/worsen or no significant change; Likelihood ratio Chi square, $p=0.0006$).”

By limiting the analysis to cells that significantly improved or significantly worsened, we also found a significant group difference (Fisher’s exact test, two-tail, $p=0.0002$).

13. What might be the social cues that induce changes in LFP? Have the authors tried to record vocalizations?

In a previous report, we recorded vocalizations during the social exposure sessions, and showed that 60% of the demonstrators’ vocalizations were produced around the time of trial initiation (i.e., nose-poke) and presentation of the Go or Nogo sound stimulus (Paraouty et al., 2021; Figure 4). We also showed a correlation between the mean number of time-locked demonstrator calls (within ± 0.5 s of the sound onset) and the number of days taken by the respective *social observers* to reach the criterion d' of 1.5. The larger the number of time-locked demonstrator calls, the faster the *social observer*’s rate of task acquisition was.

Unfortunately, we did not record vocalizations during exposure sessions in the current study. It is plausible that the LFP “anticipatory response” before nose-poke during late exposure for the *social observers* may be linked to the activity of the modulatory signal mediated by nucleus basalis or VTA or locus coeruleus, elicited by the demonstrator’s vocalizations or sound localization of the demonstrator’s position in the cage.

The post-reward response during late exposure for the *social observers* may also be linked to the activity of the above-mentioned modulatory nuclei, elicited by the sound of demonstrator chewing.

Since the anticipatory and reward LFP signals are similar during late exposure for the *social observers* and when the animal is actually practicing the task, it is likely to reflect a modulatory signal mediated by nucleus basalis or VTA or locus coeruleus. In principle, such signaling could lead to increased attention (NB), reward prediction (VTA), or arousal (LC).

In line with the Reviewer’s comment below (question #14), we have proceeded to further analyzing the LFP data during practice and included those additional Discussion points in Lines 505 - 514: “The *social observers*’ anticipatory response to the demonstrator’s nose-poke during late exposure may be linked to the activity of neuromodulatory signals. For example, dopaminergic signaling has been implicated in auditory social learning (Paraouty et al., 2021). In principle, dopaminergic or other neuromodulatory signals could be recruited by demonstrator-associated sounds such as vocalizations (60% of demonstrators’ vocalizations were found to be initiated around the time of trial initiation; see Figure 4 in Paraouty et al., 2021), movement-generated sounds, or the sound of demonstrator chewing. Since the anticipatory LFP signal during practice is correlated with the *social observer*’s

subsequent behavioral performance, it is likely to reflect a modulatory signal linked to increased attention (nucleus basalis), or reward prediction (VTA), or arousal (locus coeruleus).”

14. The LFP data are a little underwhelming. Have the authors tested to see if the LFP signal correlates with behavioral performance?

To address the Reviewer’s question, we have now performed additional analysis during the practice session, and added the following text to the Results section, Lines 396 - 406: “For *social observers*, during practice, a significant correlation was found between the average power of the low frequency band during the nose-poking period and the animal’s behavioral d’ (Pearson’s, $r=0.31$, $p=0.037$). In contrast, no significant correlation was found for the high frequency band ($r=0.09$; $p=0.563$). For the *non-social exposure* animals, during practice, no significant correlation was found between the average power during the nose-poking period and the animal’s behavioral d’ ($r=0.15$, $p=0.466$; $r=0.12$; $p=0.615$ for low and high frequency bands, respectively). For both groups, no significant correlation was found between the average decrease in power during the reward period and the animal’s behavioral d’ (for *social observers*: $r=0.07$, $p=0.724$; $r=0.04$, $p=0.855$ for low and high frequency bands, respectively; for *non-social exposure* animals: $r=0.03$; $p = 0.890$; $r=0.09$; $p=0.563$). Together, these results indicate that the induced LFP activity during nose-poke can predict the behavioral performance during practice sessions for *social observers* only.”

We have also added the following Discussion points, Lines 512 - 514: “Since the anticipatory LFP signal during practice is correlated with the *social observer*’s subsequent behavioral performance, it is likely to reflect a modulatory signal linked to increased attention (nucleus basalis), or reward prediction (VTA), or arousal (locus coeruleus).”

Reviewer #2 (Remarks to the Author):

In this study, Parouty et al. investigated the role of auditory cortex (AC) plasticity for task acquisition during social learning. To this end, the authors utilized a social learning paradigm previously established in the same lab that consists of a demonstrator and an observer animal and record from or transiently inactivate AC. The authors provide convincing evidence that AC plasticity in observers improves neural sensitivity to task relevant metrics, which is correlated with social exposure to the demonstrator (i.e., improvements are not present after passive cue exposure). They further report that this AC plasticity supports subsequent behavioral performance / task acquisition. In line with these results, inactivation of AC of observers delayed task acquisition. Finally, LFP analyses are presented demonstrating that changes in the dominant frequency bands in AC of experienced observers are induced by social cues. Together, the authors thus provide novel insight how sensory information during social learning alters neural coding in AC and how this plasticity helps subsequent learning of the observed task.

The results are presented in a highly comprehensible and elegant fashion that make the paper a joy to read. The findings are explained very clearly and discussed coherently in context of the current literature of social learning and cortical plasticity, and the conclusions are consequently well supported and convincing. The work will be of great interest to neuroscientists interested in sensory processing and (social) learning and will represent an important reference for future studies in this field.

Specific comments

I only have a few comments for clarification purposes.

1. Line 184: Can you really claim “causality” in terms of social learning at this stage? One might argue that the transient inactivation is somewhat too unspecific, as it could also suppress any processing and plasticity not directly related to social learning, and might rather induce general learning or acquisition deficits of some sorts. To claim specific causality, I would recommend testing AC inactivation in non-social exposed animals to control if no delay in task acquisition occurs compared to a saline-infused non-social exposed group.

This comment is complementary to Reviewer #1’s first three comments, and suggests the same experiment (this review even specifies the experimental result that would allow us to conclude that muscimol blocked a social signal). Taken together, these comments suggest that we need to be far more cautious in our interpretation. However, the primary reason for doing the experiment was to determine whether AC was a reasonable location from which to record, and we believe that logic still holds. We have toned down our original conclusion in the Results section, Lines 184-186: “These results indicate that the AC may contribute to auditory social learning, and suggests that it may store task-specific acoustic information during the exposure phase.”

2. Line 246: Please give exact p values (instead of “ $p>0.05$ ”). Variability is quite high for the practice groups, and it would be interesting to know if the p values indicate a trend, even if not reaching the selected alpha criterion.

The exact p values are now indicated in Lines 257 -259: “Although there is a trend, no significant group difference was found during early exposure ($p=0.083$) and during practice ($p=0.055$).”

The Reviewer is correct that there might be a trend for higher VS values for the *non-social exposure* animals during practice. Hence, we concluded in Lines 260-262: “Together, these results suggest that both social and non-social exposure are associated with changes in AC neuron firing rate and temporal processing.”

3. Line 266: How did you assess the identity of neurons across sessions? I.e., how did you make sure it is (or is not) the same unit across days? I could only find information about the general sorting procedure in the Methods.

We have now detailed how we identified single units across exposure sessions in the Results Section, Lines 282 -285 and in the Methods Section, Lines 798 -801: “Figure 3g included a subset of single unit data recorded from the same animal across exposure days, from the same general recording area (no advancement in probe across exposure days), from the same specific electrode site (63 recording sites), with similar spike waveform (visual comparison) and with similar spontaneous firing rate (± 2 Hz; mean = 0.77 ± 0.89 Hz).” We have also added the word “putative” to our description of single-units held across exposure sessions in the manuscript (Line 289) and in the figure legend (Line 592).

4. Line 396: I agree that this represents a major finding. However, it should be noted that significant plasticity (in terms of altered rate, CoeffV, and VS) also occurred prior to practice in the non-social exposure group (Fig 3.b). Panels 3f and 3g indicate that these changes might be less “advantageous” for decoding, but nonetheless they suggest some sort of improvement.

We have now clarified this point in Lines 437 - 440: “Our results showed that both types of exposure altered basic AC response properties to task-relevant cues (Figure 3b-d). However, only social exposure led to an improvement in AC single neuron and population sensitivity to the task cues, as compared to non-social exposure (Figure 3g-h, Figure 4a-c).”

5. Figure 3: It took me some time to figure out the color code (shades of blue and red), since the respective labels are positioned at the bottom of panel b of a very busy figure. Maybe one can think about another / additional way to convey the color code?

We have updated Figure 3 by simplifying the color shading scheme. We now use one shade of blue or orange for Exposure sessions, and darker shades for Practice sessions (i.e., we no longer use separate shades for Early vs. Late Exposure). In addition, labels are now provided for each panel. For consistency, this color scheme was also applied to Figure 5, Supplementary Figure 5, and Supplementary Figure 7.

6. Figure 4: It somewhat struck me that all metrics shown in panes a – d exhibit a rather large difference between the last day of exposure and start of practice for the social observers, but not for the non-social group. How do the authors interpret / explain this social-specific discrepancy? I would have expected a continuation of each metrics’ values from exposure to practice, not the least because the authors made the argument that the change in the metrics during exposure might be beneficial for task acquisition.

We agree with the Reviewer that if the improvement in AC neuron sensitivity during exposure reflects a memory that is *used* during practice, then d’ would be expected to remain high for *social observers*.

To address this interesting question, we have added the following paragraph to the Discussion section, Lines 482 - 497: “If the improvement in AC neuron sensitivity during exposure reflects a memory that is used during practice, then neural d' would be expected to remain high at the onset of practice for *social observers*. In contrast, we found a decrease in neural d' at the onset of practice (Figure 4). One plausible explanation is that AC network plasticity acquired during exposure is present but is not initially observed during task performance because it must first become associated with the top-down circuitry that supports task acquisition. Thus, the enhanced neural sensitivity (i.e., ability of AC neurons to discriminate between the Go and Nogo stimuli) induced by social exposure would have to be integrated with new information about task initiation (i.e., nose-poking to initiate sounds) and task structure (i.e., availability of reward following Go stimuli in a limited time window, and avoidance of food tray following a Nogo stimuli to prevent time-outs). Such task-dependent modulation of neural coding properties is common. For example, a large difference in AC or parietal cortex neuron sensitivity is observed when switching from passive listening to task engagement, even in fully trained animals (e.g., von Trapp et al., 2016; Caras and Sanes, 2017; Yao et al., 2023). Therefore, the variables associated with task engagement during practice may temporarily dominate the previously learned signal. Further analysis on the geometric measures of dimensionality and radius of manifolds, as well as the firing rate profile and coding sparsity of neurons within each manifold should offer a better understanding of such network reorganization.”

Reviewer #3 (Remarks to the Author):

The manuscript by Parouty et al. entitled ‘Sensory cortex plasticity 1 supports auditory social learning’, explores the role of the auditory cortex (AC) in social learning of an auditory discrimination, as well as the changes in AC neuronal activity associated with this learning. The authors use a social learning paradigm previously established by the authors (Parouty et al., 2020 and 2021), where they demonstrated that gerbils exposed to the sounds and smells of a demonstrator performing an auditory go/no-go discrimination task, subsequently learned the task faster than gerbils that had only been exposed to subsets of the task parameters not including the demonstrator. Here the authors exposed two groups of gerbils. The social observer group underwent social exposure, while the non-social exposure group was exposed for the same amount of time to the sounds of the experimenter performing the task. All gerbils were then trained in the task. The social observers learnt the task faster than the non-socially exposed gerbils. The authors tested the hypotheses that the auditory cortex plays a role in social learning of an auditory discrimination and that social learning enhances neuronal sensitivity to task-related sounds, finding evidence for both. Muscimol inactivation of the AC during social exposure led to a decrease in the exposure-induced speed of subsequent learning indicating that the AC contributed to this facilitation. Recordings from the AC during social and non-social exposure and subsequent training, revealed that while both social and non-social exposure led to improved temporal precision of sound representation, only social exposure led to an increase in neuronal discrimination, measured through neuronal d' , and an increase in neuronal classification and representation stability, measured using MFT manifold analysis. The study is a very important and elegant contribution to the understudied mechanisms of social learning. It is well designed, uses sufficient subject numbers and it is generally appropriately controlled.

Some improvements are suggested below:

Specific comments

1. The authors use, throughout the introduction and the discussion, examples of vocalization learning. To me and a priori, there is a big difference between social learning of non-social tasks and the learning of social behaviors such as communication. Since there are plenty of examples for social learning of non-social tasks and this is what is being tested in this work, I think it would be appropriate to restrict examples to this domain. The potential distinction of lack of it between social learning of social and non-social behaviors could, and should, nonetheless be included in the text.

In the Introduction, our definition of social learning has been adopted from the literature on this subject: “It typically occurs when a naïve animal experiences a conspecific performing a well-defined behavior and, as a consequence, acquires that behavior more rapidly than would occur without the social element (Bandura, 1971; Zentall, 2012; Gariépy et al., 2014; Carcea and Froemke, 2019).” Thus, it includes learning of both social and non-social tasks. However, we take the Reviewer’s point that this difference should be more accurately presented in the manuscript. Rather than restrict the examples to only learning of non-social tasks, we have added text to the Discussion section that expands on this difference and clearly defines our paradigm as one in which a non-social task is being learned. We leave open the possibility that an overlapping set of neural mechanisms contribute to the acquisition of both social and non-social tasks.

We have added the following text to the Discussion section, Lines 523 - 526: “While there may be differences between social learning of social tasks (e.g., vocalization or speech acquisition) vs. non-social tasks (as examined in the current study), we cannot exclude the possibility that an overlapping set of neural mechanisms contribute to the acquisition of both social and non-social tasks.”

2. The comparison with practice performance in the absence of social exposure is missing at points (see comments below) and would be useful. In general, going through the different non-social and social (with yoked reward, for example) exposures used by the authors, one cannot help but wondering whether there is something categorically different about the social aspect. Alternatively, and equally interesting, it could be that exposure learning is cumulative and more efficient the more information is provided, something that is maximized when a social demonstrator is present. I would appreciate a discussion of this point in the paper with evidence for and against.

Despite all the different behavioral paradigms we have tested in Paraouty et al., 2020 and 2021 (see Supplementary Figure 4 of Paraouty et al., 2021 for a summary), we cannot say that social learning is categorically different from other forms of learning. Our results indicate that non-social exposure does facilitate learning, albeit to a lesser extent than social exposure. Our sense is that a satisfactory answer to this question will depend on whether or not the social learning mechanism(s) are identical to those found for non-social forms of learning at the cellular and network level.

Similarly, we cannot say that social learning is cumulative and more efficient with more information. Access to both visual and auditory cues (Paraouty et al., 2020 - Figure 1) do not lead to faster learning as compared to access to auditory cues only (Paraouty et al., 2020 - Figure 3). This is not due to a ceiling effect (in other words, a best learning rate possible for this task) as the yoked reward condition (Paraouty et al., 2021 - Figure 1) and the dopamine (D1/D5) agonist (Paraouty et al., 2021 - Figure 3) both led to faster learning.

In this paper, we decided to focus on the neural changes during social learning and we chose to use as control condition, the non-social exposure. This control was chosen as we wanted to examine whether learning in the social vs. the non-social exposure condition led to contrasting changes in the auditory cortex. We have clarified this in the first Discussion paragraph, Lines 437 - 440: “Our results showed that both types of exposure altered basic AC response properties to task-relevant cues (Figure 3b-d). However, only social exposure led to an improvement in AC single neuron and population sensitivity to the task cues, as compared to non-social exposure (Figure 3g-h, Figure 4a-c).”

3. In relation to the previous point, is the quality of the learning in the social and non-social exposure groups different? It seems that some aspects of the LFP changes that are present during practice in the social group are not present in the non-social exposure group. Some information about the differences, beyond the presence of a gerbil, between the social and non-social exposure patterns would help in this direction.

Indeed, there are several neural measures that indicate a difference in the quality of learning following non-social exposure. In Figures 4a-c, we find that d' , capacity, and radius change more slowly during practice for the *non-social exposure* animals, as compared to the *social observers*.

The comment regarding LFP during practice is in line with Reviewer #1 (question #14), and we have now performed additional analysis during the practice sessions, and added the following text in the Results section, Lines 396 - 406: “For *social observers*, during practice, a significant correlation was

found between the average power of the low frequency band during the nose-poking period and the animal's behavioral d' (Pearson's, $r=0.31$, $p=0.037$). In contrast, no significant correlation was found for the high frequency band ($r=0.09$; $p=0.563$). For the *non-social exposure* animals, during practice, no significant correlation was found between the average power during the nose-poking period and the animal's behavioral d' ($r=0.15$, $p=0.466$; $r=0.12$; $p=0.615$ for low and high frequency bands, respectively). For both groups, no significant correlation was found between the average decrease in power during the reward period and the animal's behavioral d' (for *social observers*: $r=0.07$, $p=0.724$; $r=0.04$, $p=0.855$ for low and high frequency bands, respectively; for *non-social exposure* animals: $r=0.03$; $p=0.890$; $r=0.09$; $p=0.563$). Together, these results indicate that the induced LFP activity during nose-poke can predict the behavioral performance during practice sessions for *social observers* only.”

We have also added the following Discussion points, Lines 505 - 514: “The *social observers*' anticipatory response to the demonstrator's nose-poke during late exposure may be linked to the activity of neuromodulatory signals. For example, dopaminergic signaling has been implicated in auditory social learning (Tanaka et al., 2018; Paraouty et al., 2021). In principle, dopaminergic or other neuromodulatory signals could be recruited by demonstrator-associated sounds such as vocalizations (60% of demonstrators' vocalizations were found to be initiated around the time of trial initiation; see Figure 4 in Paraouty et al., 2021), or movement-generated sounds, or the sound of demonstrator chewing. Since the anticipatory LFP signal during practice is correlated with the *social observer*'s subsequent behavioral performance, it is likely to reflect a modulatory signal linked to increased attention (nucleus basalis), or reward prediction (VTA), or arousal (locus coeruleus).”

4. The analysis of the encoding of social cues is important, although a lack of modulation by specific social cues would not diminish the importance of the findings of increased neuronal sensitivity to task-related sounds under social exposure. The transition to LFP analysis and the choice of frequency bands is a bit sudden. Some guidance through the reasons for this transition would be welcomed.

This point is covered in the section below related to Social Cues, numbered 25.

5. Statistics in general: I would suggest the use of linear models, or at least take into account repeated measures and multiple comparisons.

We have reviewed and updated all our statistical comparisons, and added the following details in the Supplementary Information section, Lines 1134 - 1193.

“For Figure 1:

To assess the impact of muscimol infusions, we first used Kruskal-Wallis rank sum test to compare the number of days to criterion d' for the following groups, followed by post hoc nonparametric comparisons for all pairs with the Steel-Dwass method:

- Saline-infused *social observers*,
- Muscimol-infused *social observers*,
- Non-visual *social observers* (from Figure 3 in Paraouty et al., 2020),
- Muscimol-infused *social observers* - 1 day,
- Muscimol-infused *social observers* - 2 days.

Next, to compare the performance of muscimol-infused *social observers* with all our previous control paradigms (from Paraouty et al., 2020), we used the same approach described above, and compared the following groups:

- Muscimol-infused *social observers*,
- Controls for all exposure (from Figure 2c in Paraouty et al., 2020),
- Controls for social exposure (from Figure 2a in Paraouty et al., 2020),
- Controls for cage exposure (from Figure 2b in Paraouty et al., 2020).

For Figure 2:

To assess the impact of social vs. non-social exposure, as well as electrode implantation, we first used Kruskal-Wallis rank sum test to compare the number of days to criterion d' for the following groups, followed by post hoc nonparametric comparisons for all pairs with the Steel-Dwass method:

- Electrode-implanted *social observers*,
- Electrode-implanted *non-social exposure* animals,
- Non-implanted *social observers* (from Figure 3 in Paraouty et al., 2020),
- Non-implanted *non-social exposure* animals (from Figure 4b in Paraouty et al., 2020).

For Figure 3:

To compare Firing rate, we used mixed-model ANOVA's followed by post hoc Tukey-Kramer HSD comparisons with corrected alpha values.

Groups: social observers vs. non-social exposure animals.

Epochs: Early exposure, Late exposure and Practice.

Firing rate variables: Spontaneous, Unmodulated noise, and Modulated noise.

To compare the coefficient of variation, we first used Kruskal-Wallis test, followed by post hoc nonparametric comparisons for all pairs with the Steel-Dwass method. The same method was used for vector strength comparisons.

To compare the neural d' for all single units, we first used Kruskal-Wallis test, followed by post hoc nonparametric comparisons for all pairs with the Steel-Dwass method.

To compare the neural d' for single units from the same recording sites, we used two-sample t-tests to test for significant differences, and computed a Likelihood ratio Chi square to assess the changes in the 2 groups.

We calculated Pearson's r and its associated p value to analyze the relationship between median change in neural d' for all single units and days to reach behavioral criterion d' .

For Figure 4:

To look at population neurometrics, we computed mixed-model ANOVA's for each of the following variables:

- SVM neural d' ,
- Manifold capacity,
- Manifold radius,
- Manifold dimension.

For Figure 5:

For each individual animal, we used two-samples t-tests to analyze the LFP average differences as compared to baseline in the 2 regions of interest:

- Poke, and

- Reward.”

Abstract

6. Line 59: I am not sure that ‘human aureal communication’ is the best example given the focus of the manuscript. Human aureal communication is learned through social interaction but it is not a behavior that is learned just by observation. Efficient learning in infants depends on the communication being directed at them. While it undoubtedly has components of social learning, it might be a special case because it is, itself, a social behavior. The number of youtube videos demonstrating all sorts of behaviors, from cooking and knitting to putting together an engine or fixing a watch, demonstrates the extensive use of social learning.

While we accept the Reviewer’s observation that an analogy to human aural communication is limited, we are attempting to communicate the relevance of our research to as broad a group of scientists as possible. Therefore we hope to retain this text.

7. Line 63: for the naïve reader this sentence would be clearer if it were to declare when the manipulation was done and when the effect tested.

We have updated the abstract as follows, Line 60: “We first tested the prediction that AC activity is required for social learning by transiently inactivating AC during the exposure sessions only.”

8. Line 67 and 70: I am not sure what ‘improvement’ (line 67) and ‘poorer’ (line 70) means. What is the comparison?

- Improvement in the sensitivity of AC neurons with social exposure - i.e., when comparing early to late exposure for the *social observers*.
- Exposure to auditory task cues only (no social exposure) was associated with poorer neurometric and behavioral outcomes as compared to social exposure.

We have updated the abstract as follows, Lines 63 - 68: “In fact, social exposure to a performing demonstrator caused an improvement in the sensitivity of AC neurons to auditory task cues (i.e., when comparing early to late exposure sessions), as assessed with both single neuron and population metrics, and the magnitude of neural change during exposure correlated with task acquisition. In contrast, exposure to auditory task cues only was associated with poorer neurometric and behavioral outcomes, as compared to social exposure.”

9. Graphical abstract: bottom right graph, the y axis legend ‘practice acquisition rate’ is misleading. Higher rate of acquisition typically means faster learning but I assume here the higher numbers reflect not higher rate of acquisition but longer time/more trials to reach criterion? The y axis is correct in the ‘real’ graph.

We thank the Reviewer for pointing this out. We have updated the y-axis label to “Days to learn”.

Introduction

10. Second paragraph: As a reader, in order to assess whether the question of the manuscript aims to add one more example of neuronal plasticity to the list or whether it goes a step further in implicating

a specific structure and a specific mechanism in a specific social learning behavior, it would be useful to have more detail on the effects that the listed manuscripts found. What does it mean that a structure was ‘implicated’, ‘involved’, or ‘contributes’?

We have clarified in the Introduction section, Lines 108 - 109: “The social learning paradigms for which neural mechanisms are well-studied implicate a broad range of structures - through the use of both correlational and causational experimental designs.” In addition, given the brevity of the Introduction, we provide one additional sentence that describes a specific causal experiment (i.e., Jeon et al., 2010) to provide the reader with an explicit experimental example that is similar to the approach that we used here (inactivation), Lines 112- 113: “Thus, local inactivation of the anterior cingulate cortex substantially impaired observational fear learning (Jeon et al., 2010).”

For auditory researchers, we believe our study represents an addition to an already existing literature on auditory cortex plasticity. However, we would emphasize that our study explores AC plasticity during the *exposure period*, rather than during task performance as has been classically examined. For social learning researchers, our study expands the experimental paradigm from fear learning and spatial observational learning to *auditory social learning*.

For the different articles we refer to in the second Introduction paragraph, we have described below whether a correlation between neural activity and behavior was shown or a causal relationship was demonstrated with a manipulation. However, this information does not appear in the revised manuscript.

Olsson A, Phelps EA (2007) Social learning of fear. Nature neuroscience, 10(9), 1095-1102.

A review article regarding social learning of fear through observation and instruction in which correlations are discussed. “Under most circumstances, the role of the amygdala in fear conditioning is best understood together with other functional regions within a greater circuitry of fear learning. This circuitry involves sensory input and motor output systems, as well as regions that contribute to explicit and conscious aspects of learning and expression of fear.”

Jeon D, Kim S, Chetana M, Jo D, Ruley HE, Lin SY, Rabah D, Kinet JP, Shin HS (2010) Observational fear learning involves affective pain system and Cav1. 2 Ca²⁺ channels in ACC. Nature neuroscience, 13(4), 482-488.

Study in which inactivation was used to show the causal relationship between ACC and other structures in fear learning. “Inactivation of anterior cingulate cortex (ACC) and parafascicular or mediodorsal thalamic nuclei, which comprise the medial pain system representing pain affection, substantially impaired this observational fear learning, whereas inactivation of sensory thalamic nuclei had no effect. These results demonstrate the functional involvement of the affective pain system and Ca(v)1.2 channels of the ACC in observational social fear.”

Allsop SA, Wichmann R, Mills F, Burgos-Robles A, Chang CJ, Felix-Ortiz AC, Vienne A, Beyeler A, Izadmehr EM, Glober G, Cum MI, Stergiadou J, Anandalingam KK, Farris K, Namburi P, Leppla CA, Weddington JC, Nieh EH, Smith AC, Ba D, Brown EN, Tye KM (2018) Corticoamygdala transfer of socially derived information gates observational learning. Cell 173:1329-1342.

Study where targeted projections were inactivated to show the causal relationship between ACC and other structures and fear learning. “Inhibition of ACC→BLA alters real-time amygdala representation of the aversive cue during observational conditioning. Selective inhibition of the ACC→BLA projection impaired acquisition, but not expression, of observational fear conditioning.”

Leggio MG, Molinari M, Neri P, Graziano A, Mandolesi L, Petrosini L (2000) Representation of actions in rats: the role of cerebellum in learning spatial performances by observation. Proceedings of the National Academy of Sciences, 97(5), 2320-2325.

Study where lesions were used to show the causal relationship between the cerebellum and observational learning of the Morris water maze task. “After this observation training, “observer” rats underwent a hemi cerebellectomy and then were tested in the Morris water maze. In spite of the cerebellar lesion, they displayed no spatial defects, exhibiting exploration abilities comparable to controls. When the cerebellar lesion preceded observation training, a complete lack of spatial observational learning was observed.”

Petrosini L, Graziano A, Mandolesi L, Neri P, Molinari M, Leggio MG (2003) Watch how to do it! New advances in learning by observation. Brain Res Rev 42:252-264.

Review article examining both correlational and causational findings regarding the role of cerebellum in spatial observational learning. “The finding that the cerebellum is involved in procedural acquisition and in observational learning allowed us to dissect a complex behavior into single behavioral units forming a complete procedural sequence, demonstrating that such behavioral units do exist and can be independently acquired.”

Basham ME, Nordeen EJ, and Nordeen KW (1996) Blockade of NMDA receptors in the anterior forebrain impairs sensory acquisition in the zebra finch (*Poephila guttata*). Neurobiol Learn Mem 66:295-304.

Study using causational design to better understand the neural mechanisms of vocal learning. “To test the hypothesis that sensory acquisition requires activation of NMDA receptors in or near the IMAN we infused the NMDA receptor antagonist directly into the anterior forebrain.”

Phan ML, Pytte CL, Vicario DS (2006) Early auditory experience generates long-lasting memories that may subserve vocal learning in songbirds. PNAS, 103(4), 1088-1093.

Study using correlations to better understand the neural mechanisms of vocal learning. “We show that neurons in a forebrain auditory area of adult male zebra finches are selectively tuned to the song of a tutor heard early in development. Furthermore, the strength of this selectivity shows a striking correlation with the fidelity of vocal imitation, suggesting that this auditory memory may have served as the model for song learning.”

London SE, Clayton DF (2008) Functional identification of sensory mechanisms required for developmental song learning. Nat Neurosci, 11(5), 579-586.

Study with causational design to investigate the role of a brain structure in vocal learning. “Here we test the hypothesis that molecular signaling in a sensory brain area outside of the song system is required for developmental song learning. Using controlled tutoring and a pharmacological inhibitor, we transiently suppressed the extracellular signal-regulated kinase signaling pathway in a portion of the auditory forebrain specifically during tutor song exposure.”

Yanagihara S, Yazaki-Sugiyama Y (2019) Social interaction with a tutor modulates responsiveness of specific auditory neurons in juvenile zebra finches. Behav. Processes. 163, 32–36.

Study using correlations and differential effects to better understand the neural mechanisms of vocal learning. “Recently we found a subset of neurons in the higher-level auditory cortex of juvenile zebra finches that exhibit highly selective auditory responses to the tutor song after song learning, suggesting an auditory memory trace of the tutor song (Yanagihara and Yazaki-Sugiyama, 2016). Here we show that auditory responses of these selective neurons became greater when juveniles were paired with their tutors, while responses of non-selective neurons did not change.”

Pagliaro AH, Arya P, Piristine HC, Lord JS, Gobes SMH (2020) Bilateral brain activity in auditory regions is necessary for successful vocal learning in songbirds. Neurosci Lett, 718, 134730.

Study where inactivation of brain regions were used to show the causal relationship with vocal learning. “Here, we hypothesize that inhibition of the left NCM during interaction with a song tutor would impair imitation of the tutor's song more than inhibition of the right NCM.”

Mooney R (2009) Neural mechanisms for learned birdsong. *Learn. Mem.* 16, 655–669.

Review article examining both correlational and causational findings regarding songbird vocal learning.

11. Line 121: ‘Therefore, social experience may have a general effect on sensory encoding of environmental cues, and this initial plasticity may facilitate the acquisition of new behaviors.’ I think the concept exposed here is the crux of the matter and could be explained a bit more extensively. What is it meant by ‘a general effect on sensory encoding environmental cues’?

We agree with the reviewer that this definition is important to clarify. We have updated the following text to make this point clear, Lines 119 - 122: “Therefore, social experience can improve sensory encoding of environmental cues (i.e., neuronal responses become more selective for relevant stimuli), and this initial plasticity may facilitate the acquisition of new behaviors.”

12. Line 133: You could include de Hoz et al., *Cerebral Cortex* 2018 PMID: 28334281, where early gene expression specifically in AC was shown to be important in discrimination learning.

We have now included this reference as suggested by the reviewer.

Results

13. Role of AC – muscimol:

Exposure under muscimol: was there any behavioral difference between saline- and muscimol-injected animals during the exposure phase?

In response to Reviewer #1 comment along the same lines (question #1), we have added the following to the Methods Section of the manuscript, Lines 738-741: “Prior to starting the social exposure session with a demonstrator, all infused animals were monitored for a few minutes in the test cage to assess their locomotion and general behavior. No animals required a longer recovery period, and all animals were alert and engaged, and displayed proper posture and normal motor functions.”

To directly address the Reviewer’s question, all muscimol-infused animals behaved similarly to saline-infused animals during the exposure sessions. In fact, an identical dose of muscimol was used by Caras and Sanes (2017), see Supplementary Figures 10 and 11, and the authors concluded that this dose of muscimol did not grossly perturb psychometric performance in adult gerbils as they observed excellent detection of AM, no effect on trial completion rates, on false alarm rates nor on reaction times.

14. According to the methods, day 5 of exposure was directly followed with day 1 of practice. Were the muscimol animals not under the effect of muscimol on day 1 of practice? Please explain and justify the design if this is the case.

The Reviewer’s concern is in line with a comment expressed by Reviewer #1 (i.e., the longevity of muscimol’s effect on AC). Indeed, the first practice session occurred directly after the last exposure

session (i.e., day 5 of exposure). As described in our answer to Reviewer #1's (see full answer to questions #2 and #3):

To account for this possibility, we have recalculated the difference in rate of task acquisition between the muscimol- and saline-infused animals after excluding the first day(s) of practice:

- after excluding the first day of practice, we still found a significant delay in task acquisition (Steel-Dwass comparison, $p=0.033$), and
- after excluding the first 2 days of practice, we still found a significant delay in task acquisition ($p=0.049$).

This is now explicitly acknowledged in the Results section, Lines 188 - 191: "To evaluate whether the effect of muscimol infusion persisted into the first one or two days of practice, we excluded these days from the data analysis and confirmed a significant delay in rate of task acquisition, as compared to the saline-infused *social observers* (after excluding practice day 1, $p=0.033$; after excluding practice days 1 and 2, $p=0.049$)."

15. The time course of the effect of muscimol is difficult to control. How was it assessed? Do the authors have any evidence that the AC of muscimol-treated gerbils was fully functional during the first 2-3 days of training?

This point is addressed above in response to question #14.

16. Supplementary figure 1: The delay in learning is difficult to interpret without knowing how a naïve unexposed animal would behave. Based on Paraouty et al., 2020 Figure 2c, it looks like the muscimol animals perform similarly, maybe slightly better, than non-socially exposed animals. I think this is an important comparison that should be included in the current manuscript. It is important because it helps interpret the data and, although it is no proof that muscimol has not necessarily left post-exposure long lasting effects that have delayed the learning, it is consistent with it. At the same time, it looks like the delay in learning arises from a delay in making enough go trials. Once no-go trials are introduced, on day 3 for the saline groups and on day 6 for the muscimol group, the rate of learning is faster for the saline group than the muscimol group. Both groups, however, are faster than the untreated and unexposed animals in Paraouty et al., 2020 Figure 2c, which need 7 days to reach a d' of 1.5. What do the authors think is the meaning of this?

To address the Reviewer's question about correlation, we compared learning rate of the muscimol-infused *social observers* and the *non-social exposure* animals. No significant difference was found in terms of number of days to reach criterion d' (Wilcoxon rank sum test, $X^2(1)=0$, $p=1$). Despite this finding, we wish to acknowledge that the types of cues used by the animals in each condition during exposure may still be very different.

To address the Reviewer's comment regarding other control groups, we have added this text to the Results Section, Lines 191 - 198: "Furthermore, to determine whether muscimol-infused *social observers* benefited from the social exposure sessions, we compared their learning curves to previously collected control groups in which there was no social experience (Paraouty et al., 2020). These comparisons revealed that muscimol-infused *social observers* continued to display an effect of *social exposure* as they performed significantly better than all previously obtained control groups (comparison with Control for social exposure from Paraouty et al., 2020, Figure 2a: no animal

reached criterion d' ; comparison with Control for cage exposure from Figure 2b: $p=0.049$; comparison with Control for all exposure from Figure 2c: $p=0.011$)."'

17. Figure 1c and supplementary figure 4: It looks like a white rectangle was put over the figure to cover the lower values. It is very disconcerting. I think the figures should be replotted.

The white patch is to show discontinuity between the y axis "no d' " label and the rest of the y-axis with actual measures of d' .

To address the Reviewer's comment, we have changed each figure that plots behavior such that the white area is partially transparent. This has the advantage of making the figure style consistent with our two previous publications (Paraouty et al., 2020 and 2021). To make this clear, we have added this information in all concerned Figure legends: "No d' was computed when the observers initiated <15 Nogo trials during the practice session, and those points are separated from the actual d' measures (on the y-axis) by a white patch."

Neuron response properties

18. Line 232: 'In contrast, no changes in firing rate occurred during silence or during the unmodulated portion of the sound stimulus (see Supplementary Figure 5, $p>0.05$).' because all other comparison are including both exposure phases and the practice phase, this sentence leads to believe that there is no difference between all 3 phases, but the firing rate is significantly higher during the practice phase according to Sup. Fig. 5. Please expand.

We have updated this line to specify that no significant change occurs from early to late exposure. This result differs from the finding mentioned in the first sentence of the paragraph (i.e., firing rate evoked by the modulated portion of the sound stimuli increased significantly from early to late exposure).

Lines 244 -245 : "No significant changes in firing rate occurred from early to late exposure during silence or during the unmodulated portion of the sound stimulus (see Supplementary Figure 5, $p>0.05$)."

19. The ranges of firing rates increase with exposure and practice. Is there a correlation between the firing rate and the neuronal d' ? and do the neurons that improve d' over time also increase their firing rate?

To address the question regarding the correlation between the firing rate and the neuronal d' : for all single units, we found no significant correlation between spontaneous firing rate and the neural d' computed using the pattern classifier analysis (Early exposure, Pearson's, $r=0.02$, $p=0.782$; Late exposure, $r=0.07$, $p=0.308$; Practice, $r=0.025$, $p=0.623$). However, when all epochs (early exposure, late exposure, practice) were grouped together, a significant correlation was found between spontaneous firing rate and neural d' ($r=0.32$, $p<0.0001$).

To address the question regarding whether neurons that improve d' over time also increase their firing rate: for all single units, we found no significant correlation between the change in neural d' and the change in spontaneous firing rate from early to late exposure (Pearson's, $r=0.082$, $p=0.278$). In contrast, when comparing the change in neural d' and the change in spontaneous firing rate from late

exposure to practice, a significant correlation was found ($r=0.26$, $p<0.0001$). This final comparison suggests that the firing rate contributes to information used by the pattern classifier analysis to compute neural d' .

Therefore, we have added text to the Methods section, Lines 825 - 828: "Comparison between changes in neural d' obtained by the spike pattern classifier and changes in spontaneous firing rate from exposure to practice ($r=0.26$, $p<0.0001$) suggest that firing rate contributes to information used by the spike pattern classifier analysis in computing neural d' ."

Neuronal sensitivity

20. One problem I see with the neural d' is that it doesn't consider the quality of the classification. For example, a spike train during a go trial might be classified correctly as a hit because it is less distant from the go template than from the no-go template and yet be quite different from both. Please include some quantification of distribution of Euclidean distances and, if applicable, describe the distance threshold for inclusion of spike trains.

If we understand correctly, the Reviewer would like us to look at the distribution of each test versus template comparison to see how "far" that value is to each template. Since the test and template trials randomly change across each iteration, we will obtain different metrics for each iteration, as well as each time we run the Classifier analysis on an individual cell. We are not clear regarding the "distance threshold" the reviewer is referring to. We have no other neural information with which to correlate the animal's behavioral decisions (Go: hit vs miss; Nogo: false-alarm vs correct-reject). Thus, our strategy with the spike train classification was to make it binary in a way that it could only match one of the two AM rates and correlate it with behavioral outcome. Since the template and test trials are randomly selected with each iteration, the Euclidean distance measures change across each iteration. However, by performing many iterations, we obtain a measure that fits with our classification strategy. Appropriately quantifying the neural classification would require a classification strategy that operates along many dimensions. This is a goal of a follow up study by one of the co-authors (JY) who developed and used this technique previously (Yao and Sanes, 2021).

21. Paragraph starting on line 265: it is notoriously difficult to demonstrate that the same single unit is being recorded from one day to the next. If the authors are arguing that they recorded from 91 and 86 single units in the social and non-social exposure groups, respectively, across the early and late phases, it would be important to show some quantification of the stability of the spike shapes and the tuning, as well as some example spike shape means across channels for both the early and late phase.

We fully agree with the Reviewer regarding how difficult it is to prove that the recordings belong to the same unit. Our reasoning for this analysis was as follows: Even if the units are not *strictly* the same, comparing units with the same characteristics (such as waveform and spontaneous firing rate) will give us a better idea of the changes occurring from early to late exposure in terms of neural d' in addition to assessing the global changes for all single units that we recorded from (see Figure 3f for all single-units; and Figure 3h for all single-units per animal). Therefore, we used a similar approach as in Caras and Sanes, 2017 (see Figure 1).

To address the Reviewer's request for quantification of stability, we have now detailed how we identified single units across exposure sessions in the Methods Section, Lines 798 -801: "For Figure 3g, we identified single-unit data recorded from the same animal across exposure days, from the same

general recording area (no advancement in probe across exposure days), from the same specific electrode site (63 recording sites), with similar spike waveform (visual comparison), and with similar spontaneous firing rate (± 2 Hz; mean = 0.77 ± 0.89 Hz).”

We have also rephrased Lines 282 - 285 in the Results Section: “To further assess whether the sensitivity of individual AC neurons improved during exposure, we restricted our analysis to a subset of single units obtained from identical recording sites, with similar waveforms and spontaneous firing rates (± 2 Hz) during both early and late exposure sessions (see Methods).”

All this being said, we have added the word “putative” to our description of single units held across exposure sessions in the manuscript (Line 289) and in the figure legend (Line 592).

22. Figure 3h and paragraph starting line 271. The plot is beautiful, yet it could be that the two groups behave differently. While it looks like there is a clear interaction between neuronal sensitivity and behavioral performance, it also seems that the correlation that is clearly observed within the social-exposure group might not be as strong in the non-social exposure group, where the neuronal sensitivity is relatively homogeneous across behavioral performance. Please discuss. Also, please explain the x axis better. I assume it is plotting, for each gerbil, the median, across all units recorded, of the neural d' difference between early vs late.

The Figure legend has been updated to explain the axes better.

Lines 597 - 599 : “Significant correlation between difference in median neural d' from early to late exposure of all single units for each animal and the number of practice days required for the given animal to perform at criterion d' .”

In addition, the Results Section now includes the statistics for each of the groups and mentions that the relationship between median change in neural d' and number of days to reach criterion d' may be different for the 2 groups. The absence of significant correlation for the non-social exposure group might be due to the small n in this group.

Lines 302 - 306: “Although a significant correlation is found for all animals (both *social observers* and *non-social exposure* animals), the exact relationship between median change in neural d' from early to late exposure and the number of days to perform at criterion d' may be different for the 2 groups (*social observers*, $n=6$, $p= 0.02$; *non-social exposure* animals, $n=4$, $p= 0.14$).”

23. Line 292; please include an intuition in the text of the MFT manifold analysis, including the capacity and, later (line 306), the radius. This is partly done in the discussion but should be included and expanded in the results.

This Results Section has been updated to include a longer Introduction to the different geometric measures, Lines 322 – 324: “To assess the high-dimensional geometry of the population code, we examined a second population metric based on the mean-field theoretic manifold analysis technique that connects stimulus encoding efficiency to decoding capacity (see Methods; Chung et al., 2018; Cohen et al., 2020; Stephenson et al. 2019)”, and in Lines 337 – 339 “In this framework, geometric measures such as radius and dimensionality inform us about the encoding of high-level features, including the extent and shape of neural variability in response to the same stimulus (Chung et al., 2018).”

24. Paragraph starting Line 314. Please refer to the subplots in Sup. Fig. 6

This paragraph has been updated to refer exactly to each of the subplots of Supplementary Figure 6.

Lines 347 - 355: “Neural population d' during exposure was significantly correlated with behavioral measures of d' during practice for all animals (Supplementary Figure 6a; $r=0.37$, $p=0.002$). Similarly, there was a correlation between manifold capacity (Supplementary Figure 6b; $r=0.49$, $p=0.002$), and manifold radius (Supplementary Figure 6c; $r=-0.40$, $p=0.012$) during exposure and the subsequent behavioral d' for all animals. Manifold dimension did not display a correlation with behavior (Supplementary Figure 6d; $r=-0.13$, $p=0.43$). In addition, neural population d' during practice was significantly correlated with behavioral measures of d' during practice for all animals (Supplementary Figure 6e; $r=0.63$, $p<0.0001$), manifold capacity (Supplementary Figure 6f; $r=0.39$, $p=0.007$), manifold radius (Supplementary Figure 6g; $r=-0.48$, $p=0.0007$), and manifold dimension (Supplementary Figure 6h; $r=0.39$, $p=0.007$).”

Social cues

25. Line 332: the transition from spikes to LFP is a bit sudden and would benefit from a short justification. Same for the use of the two frequency bands.

We have now included a longer introductory paragraph in the Results Section.

Lines 363 - 378: “To address whether AC activity was modulated by social cues from the demonstrator during the exposure sessions, we examined the correlation between local field potentials (LFP) and two demonstrator behaviors: nose-poking to initiate trials and acquisition of a reward at the food tray. First, we measured the induced power from local field potentials (LFP, see Methods) in a subset of active recording sites for each animal. Induced power represents oscillatory activity that are not necessarily time-locked to the AM signals and are often related to a variety of higher-order cognitive functions, including internally driven representations (e.g., Bertrand and Tallon-Baudry, 2000; Jeschke et al., 2008). Thus, induced power may reflect the encoding of social cues linked to the demonstrator’s behavior during exposure sessions.

Our analysis targeted Go and hit trials, and focused on 2 time periods: (1) an anticipatory phase during which demonstrators nose-poked to initiate each trial, just prior to the onset of the sound stimuli: from -0.5 to -0.1 sec (with 0 sec representing sound onset), and (2) a reward-related phase during which the demonstrator sought and obtained a food reward: from 0 to 2 sec (with 0 sec representing reward onset). We examined induced power during those 2 time periods and looked at two distinct frequency bands: a low frequency band (Figure 5; 6 to 20 Hz; thick lines) and a high frequency band (20 to 80 Hz; thin lines; see Methods).”

We have also included in the Methods section (Lines 904 - 906) a short justification for the use of the 2 frequency bands (see answer to question #26).

26. Regarding the frequency bands, theta (somewhere between 6 and 12Hz, depending on literature) is associated with locomotion. I am not sure to what extent one would detect theta in AC, but it wouldn’t be surprising given its proximity with hippocampus and the fact your electrodes are quite deep in AC. During practice one would expect an increase in theta in the approach to the reward. Would a further

dissociation of the low frequency band answer whether the anticipatory increase is locomotion related?

We initially divided our LFP data into 5 frequency bands, and did not find an elevation of the 6-12 Hz signal before reward delivery (i.e., when the animal moves from the nose-port to the food tray) as animals were actively engaged in the task during practice sessions. In the current study, in order to avoid choosing specific frequency band cutoffs and over-interpreting our LFP results, we decided to only present a low and a high frequency band as the general trends for all low (or high) frequency bands were roughly similar.

We have clarified this in the Methods section, Lines 904 - 906: “A finer decomposition of frequency bands into 5 distinct bands (6-8 Hz, 6-12 Hz, 12-20 Hz, 20-40 Hz, 40-80 Hz) revealed grossly similar results in the first three bands and the last two bands. We thus chose to merge the first three bands and the last two bands.”

27. How do the exposed social observers and the non-social exposure animals behave during exposure phase? Do they follow the demonstrator/ experimenter?

In our previous reports (Paraouty et al., 2020 and 2021), we have included a short video of such a social exposure session (see Supplementary Video S1), and we also analyzed the videos of social exposure sessions (see Supplementary Figure 2).

In the current study, we have not recorded exposure session videos, but we should note that the behavior of both *social observers* and *non-social exposure* animals are similar to those observed in Paraouty et al., 2020 and 2021. *Social observers* tend to explore the test cage and their head orientation is quite biased towards the direction of the food tray (although no food tray is present on their side of the cage). For *non-social exposure* animals, the experimenter triggered trials from outside the test booth, so there was no animal nor human inside the test booth. We should also like to note that *social observers* do not tend to strictly follow the demonstrator’s movement (few anecdotal evidence of such following behavior was observed and all occurred in the “yoked reward experiment” of Paraouty et al., 2021).

28. Figure 5: the colours are very pastel and the contrast, difficult. I suggest making the poke-bar more distinctive, so that the timing of events in the power trace relative to the poke is easier to see. I also suggest making power traces darker and thicker and to use as much of the width as is possible. The schematic of the sound could represent a lower AM... etc, etc... Just to give you an idea, I have the pdf at 150%, glasses on, and I am still struggling.

To address the Reviewer’s comment, we have added a darker outline to the poke bar to make it more distinctive. The 12-Hz AM stimulus is now also darker in color. We kept the AM rate the same because the number of cycles shown matches the time axis. These changes have been implemented for both Figure 5 and Supplementary Figure 7.

29. Why is the anticipatory increase before the poke not observed in the non-social exposure group during practice?

This question was also raised by Reviewer #1 (see our answer to Reviewer 1, question #13). To reiterate a portion of our answer, we have included this paragraph in the Discussion section, Lines 505

- 514: “The *social observers*’ anticipatory response to the demonstrator’s nose-poke during late exposure may be linked to the activity of neuromodulatory signals. For example, dopaminergic signaling has been implicated in auditory social learning (Paraouty et al., 2021). In principle, dopaminergic or other neuromodulatory signals could reflect demonstrator-associated sounds such as vocalizations (60% of demonstrators’ vocalizations were found to be initiated around the time of trial initiation; see Figure 4 in Paraouty et al., 2021), or movement-generated sounds, or the sound of demonstrator chewing. Since the anticipatory LFP signal during practice is correlated with the *social observer*’s subsequent behavioral performance, it is likely to reflect a modulatory signal linked to increased attention (nucleus basalis), or reward prediction (VTA), or arousal (locus coeruleus).”

30. Please make it explicit in text and/or figure legend that the practice plot only includes sessions with d' above 1.5. It is an important piece of information, but I had to look for the one sentence in the methods.

We have now added this information in all Figure Legends concerned: “No d' was computed when the observers initiated <15 Nogo trials, and those points are separated from the actual d' measures (on the y-axis) by a white patch.”

31. Sup. Fig. 7a-b, the y axis values are cut or wrong. The plots are very small! Have pity.

We have updated the y-axis.

32. Sup. Fig. 7c, please add the color legend to the plot.

The Figure has been updated to clearly distinguish between the different epochs.

33. What is the peak at about 8Hz that is observed in most traces in both groups and both trial types?

It is possible that the peak observed around 8 Hz is due to exploratory whisking or respiratory activity of the animals. There are reports that olfactory and somatosensory cortices send functional projections to the AC.

Discussion

34. Line 438: Please specify the connections between AC and other structures and specifically name those that are speculated to subserve social learning.

We have rephrased this sentence to include a few potential structures that may be implicated in social learning, and added two additional references in Lines 499 - 503: “Given the dense, reciprocal connections between AC and other sensory and association cortices (e.g., visual, parietal, frontal cortices, striatum, hippocampus; Budinger et al., 2000; 2006; 2008; 2009; Omer et al., 2018; Carcea and Froemke, 2019), we predicted that the AC integrates both task-specific auditory cues and relevant social cues during exposure, thereby facilitating subsequent task performance.”

35. Line 442: ‘Specifically, social information linked to demonstrator trial initiation and reward elicited significant changes in social observers AC activity.’ While likely, strictly speaking there is no evidence that what triggers changes in activity around trial initiation and reward is social information. There might be other sounds that associated with the demonstrator’s presence that, if replicated by the

experimenter would lead to the same effect. Please rephrase here and other instances. See general comments.

We agree with the Reviewer, and have now updated this sentence.

Lines 504 - 505: “Specifically, significant changes in *social observers* AC activity during exposure sessions were time-locked to demonstrator trial initiation and reward periods.”

36. Line 453 and related to previous comment, were vocalizations recorded in this study? If not, since training in previous studies was presumably identical, could the timing of social information such as vocalizations be included here and discussed in the context of LFP patterns?

We have not recorded vocalizations in the current study. However, as the Reviewer correctly points out, we have vocalization results during social exposure sessions in Paraouty et al., 2021.

This question was also raised by Reviewer #1 (see our answer to that Reviewer 1, question #13). To reiterate a portion of our answer, we have now included this paragraph in the Discussion section, Lines 508 - 512: “In principle, dopaminergic or other neuromodulatory signals could reflect demonstrator-associated sounds such as vocalizations (60% of demonstrators’ vocalizations were found to be initiated around the time of trial initiation; see Figure 4 in Paraouty et al., 2021), or movement-generated sounds, or the sound of demonstrator chewing.”

Methods

37. Housing: How were the gerbils housed? If in groups, did they have as partners gerbils from other groups (demonstrator, social exposure etc)?

In general, all gerbils (demonstrators and observers) are housed with same-sex littermates.

In all experiments of the current study, the demonstrators used were not littermates of any of the *social observers* and were not housed together with any of the *social observers*. Social learning with a cage-mate demonstrator vs. a non-cage-mate was assessed in Paraouty et al., 2020 and no significant difference was found (see Supplementary Figure 3). This is now mentioned in the Methods Section, Lines 676 - 678.

38. Non-social exposure method: to better understand whether the difference between this protocol and that for the social exposure results only from differences in the social aspect, it would be useful to have the number of nose-pokes and rewards during both exposure phases, as well as the distribution of time intervals between nose-pokes and rewards.

To address the Reviewer’s comment, we have rephrased Lines 697 - 699 in the Methods section: “In each exposure session for the *non-social exposure* gerbils, the experimenter triggered the same number of Go/Nogo trials and the same responses (i.e., hits, misses, correct rejects, and false alarms) as performed by demonstrator gerbils in the Social exposure paradigm.”

REVIEWERS' COMMENTS

Reviewer #2 (Remarks to the Author):

The authors addressed all my comments as well as the plenty additional comments by reviewers #2 & #3) with exceptional diligence and were able to improve an already excellent study.

Reviewer #3 (Remarks to the Author):

The manuscript by Parouty et al. entitled 'Sensory cortex plasticity supports auditory social learning', explores the role of the auditory cortex (AC) in social learning of an auditory discrimination, as well as the changes in AC neuronal activity associated with this learning. The authors use a social learning paradigm previously established by the authors (Parouty et al., 2020 and 2021), where they demonstrated that gerbils exposed to the sounds and smells of a demonstrator performing an auditory go/no-go discrimination task, subsequently learned the task faster than gerbils that had only been exposed to subsets of the task parameters not including the demonstrator. Here the authors exposed two groups of gerbils. The social observer group underwent social exposure, while the non-social exposure group was exposed for the same amount of time to the sounds of the experimenter performing the task. All gerbils were then trained in the task. The social observers learnt the task faster than the non-socially exposed gerbils. The authors tested the hypotheses that the auditory cortex plays a role in social learning of an auditory discrimination and that social learning enhances neuronal sensitivity to task-related sounds, finding evidence for both. Muscimol inactivation of the AC during social exposure led to a decrease in the exposure-induced speed of subsequent learning indicating that the AC contributed to this facilitation. Recordings from the AC during social and non-social exposure and subsequent training, revealed that while both social and non-social exposure led to improved temporal precision of sound representation, only social exposure led to an increase in neuronal discrimination, measured through neuronal d' , and an increase in neuronal classification and representation stability, measured using MFT manifold analysis. The study is an important and elegant contribution to the understudied mechanisms of social learning. It is well designed, uses sufficient subject numbers and it is generally appropriately controlled.

In this revision, the authors have answered my questions.

I have a couple of minor suggestions:

- 1) In Figure 1, Sup Fig 1 and 3: there is a light grey frame around the data that is distracting, especially in all panels b where it gives the impression of covering data points for animals that have either high d' values or high rates of false alarms? I would suggest removing the frame all together. I would also put the y axes close to the data, between the scheme and the plot, rather than to the left of the scheme.

2) In the calculation of performance-based d' . While making the white patch that separates performance with less than 15 Nogo trials improves the plot, the plot remains confusing. If no d' is computed, what is plotted below zero d' ? why does the semitransparent patch not cover the lowest values on the plot but only a range just below zero d' (see Figure 1b)?

I congratulate the authors for the elegant study and the beautiful set of data.

Response to Referees

Reviewer #2 (Remarks to the Author):

The authors addressed all my comments, as well as the plenty additional comments by reviewers #2 & #3) with exceptional diligence and were able to improve an already excellent study.

Thank you.

Reviewer #3 (Remarks to the Author):

The manuscript by Paraouty et al. entitled ‘Sensory cortex plasticity 1 supports auditory social learning’, explores the role of the auditory cortex (AC) in social learning of an auditory discrimination, as well as the changes in AC neuronal activity associated with this learning. The authors use a social learning paradigm previously established by the authors (Paraouty et al., 2020 and 2021), where they demonstrated that gerbils exposed to the sounds and smells of a demonstrator performing an auditory go/no-go discrimination task, subsequently learned the task faster than gerbils that had only been exposed to subsets of the task parameters not including the demonstrator. Here the authors exposed two groups of gerbils. The social observer group underwent social exposure, while the non-social exposure group was exposed for the same amount of time to the sounds of the experimenter performing the task. All gerbils were then trained in the task. The social observers learnt the task faster than the non-socially exposed gerbils. The authors tested the hypotheses that the auditory cortex plays a role in social learning of an auditory discrimination and that social learning enhances neuronal sensitivity to task-related sounds, finding evidence for both. Muscimol inactivation of the AC during social exposure led to a decrease in the exposure-induced speed of subsequent learning indicating that the AC contributed to this facilitation. Recordings from the AC during social and non-social exposure and subsequent training, revealed that while both social and non-social exposure led to improved temporal precision of sound representation, only social exposure led to an increase in neuronal discrimination, measured through neuronal d' , and an increase in neuronal classification and representation stability, measured using MFT manifold analysis. The study is an important and elegant contribution to the understudied mechanisms of social learning. It is well designed, uses sufficient subject numbers and it is generally appropriately controlled.

In this revision, the authors have answered my questions.

I have a couple of minor suggestions:

1) In Figure 1, Sup Fig 1 and 3: there is a light grey frame around the data that is distracting, especially in all panels b where it gives the impression of covering data points for animals that have either high d' values or high rates of false alarms? I would suggest removing the frame all together. I would also put the y axes close to the data, between the scheme and the plot, rather than to the left of the scheme.

Figures 1 and 2, as well as Supplementary Figures 1, 3, and 4 have been revised according to the Reviewer’s suggestion :

- The gray boxes around the performance graphs have been removed.
- The y-axis has been placed next to the performance graph.

2) In the calculation of performance-based d' . While making the white patch that separates performance with less than 15 Nogo trials improves the plot, the plot remains confusing. If no d' is computed, what is plotted below zero d' ? why does the semitransparent patch not cover the lowest values on the plot but only a range just below zero d' (see Figure 1b)?

We have modified each figure according to the Reviewer's suggestions:

- The white patch has been removed.
- All symbols and lines beneath a $d'=0$ have been removed.
- Text has been added beneath the y-axis to specify that no d' was computed when the animal performed fewer than 15 Nogo trials.

I congratulate the authors for the elegant study and the beautiful set of data.